# Stochastic Gradient Richardson-Romberg Markov Chain Monte Carlo

**Alain Durmus[1], Umut Şimşekli[1], Éric Moulines[2], Roland Badeau[1], Gaël Richard[1]**
1: LTCI, CNRS, Télécom ParisTech, Université Paris-Saclay, 75013, Paris, France
2: Centre de Mathématiques Appliquées, UMR 7641, École Polytechnique, France

## Abstract

Stochastic Gradient Markov Chain Monte Carlo (SG-MCMC) algorithms have become increasingly popular for Bayesian inference in large-scale applications. Even though these methods have proved useful in several scenarios, their performance is often limited by their bias. In this study, we propose a novel sampling algorithm that aims to reduce the bias of SG-MCMC while keeping the variance at a reasonable level. Our approach is based on a numerical sequence acceleration method, namely the Richardson-Romberg extrapolation, which simply boils down to running almost the same SG-MCMC algorithm twice in parallel with different step sizes. We illustrate our framework on the popular Stochastic Gradient Langevin Dynamics (SGLD) algorithm and propose a novel SG-MCMC algorithm referred to as Stochastic Gradient Richardson-Romberg Langevin Dynamics (SGRRLD). We provide formal theoretical analysis and show that SGRRLD is asymptotically consistent, satisfies a central limit theorem, and its non-asymptotic bias and the mean squared-error can be bounded. Our results show that SGRRLD attains higher rates of convergence than SGLD in both finite-time and asymptotically, and it achieves the theoretical accuracy of the methods that are based on higher-order integrators. We support our findings using both synthetic and real data experiments.

## 1 Introduction

Markov Chain Monte Carlo (MCMC) techniques are one of the most popular family of algorithms in Bayesian machine learning. Recently, novel MCMC schemes that are based on stochastic optimization have been proposed for scaling up Bayesian inference to large-scale applications. These so-called Stochastic Gradient MCMC (SG-MCMC) methods provide a fruitful framework for Bayesian inference, well adapted to massively parallel and distributed architecture. In this domain, a first and important attempt was made by Welling and Teh [1], where the authors combined ideas from the Unadjusted Langevin Algorithm (ULA) [2] and Stochastic Gradient Descent (SGD) [3]. They proposed a scalable MCMC framework referred to as Stochastic Gradient Langevin Dynamics (SGLD). Unlike conventional batch MCMC methods, SGLD uses subsamples of the data per iteration similar to SGD.

Several extensions of SGLD have been proposed [4–12]. Recently, in [10] it has been shown that under certain assumptions and with sufficiently large number of iterations, the bias and the mean-squared-error (MSE) of a general class of SG-MCMC methods can be bounded as $\mathcal{O}(\gamma)$ and $\mathcal{O}(\gamma^2)$, respectively, where $\gamma$ is the step size of the Euler-Maruyama integrator. The authors have also shown that these bounds can be improved by making use of higher-order integrators.

In this paper, we propose a novel SG-MCMC algorithm, called Stochastic Gradient Richardson-Romberg Langevin Dynamics (SGRRLD) that aims to reduce the bias of SGLD by applying a numerical sequence acceleration method, namely the Richardson-Romberg (RR) extrapolation, which requires running two chains with different step sizes in parallel. While reducing the bias, SGRRLD also keeps the variance of the estimator at a reasonable level by using correlated Brownian motions.

We show that the asymptotic bias and variance of SGRRLD can be bounded as $\mathcal{O}(\gamma^2)$ and $\mathcal{O}(\gamma^4)$, respectively. We also show that after $K$ iterations, our algorithm achieves a rate of convergence for the MSE of order $\mathcal{O}(K^{-4/5})$, whereas this rate for SGLD and its extensions with first-order integrators is of order $\mathcal{O}(K^{-2/3})$.

Our results show that by only using a first-order numerical integrator, the proposed approach can achieve the theoretical accuracy of methods that are based on higher-order integrators, such as the ones given in [10]. This accuracy can be improved even more by applying the RR extrapolation multiple times in a recursive manner [13]. On the other hand, since the two chains required by the RR extrapolation can be generated independently, the SGRRLD algorithm is well adapted to parallel and distributed architectures. It is also worth to note that our technique is quite generic and can be virtually applied to all the current SG-MCMC algorithms besides SGLD, provided that they satisfy rather technical weak error and ergodicity conditions.

In order to assess the performance of the proposed method, we conduct several experiments on both synthetic and real datasets. We first apply our method on a rather simple Gaussian model whose posterior distribution is analytically available and compare the performance of SGLD and SGRRLD. In this setting, we also illustrate the generality of our technique by applying the RR extrapolation on Stochastic Gradient Hamiltonian Monte Carlo (SGHMC) [6]. Then, we apply our method on a large-scale matrix factorization problem for a movie recommendation task. Numerical experiments support our theoretical results: our approach achieves improved accuracy over SGLD and SGHMC.

## 2 Preliminaries

### 2.1 Stochastic Gradient Langevin Dynamics

In MCMC, one aims at generating samples from a target probability measure $\pi$ that is known up to a multiplicative constant. Assume that $\pi$ has a density with respect to the Lebesgue measure that is still denoted by $\pi$ and given by $\pi : \theta \to \mathrm{e}^{-U(\theta)} / \int_{\mathbb{R}^d} \mathrm{e}^{-U(\tilde{\theta})} \mathrm{d}\tilde{\theta}$ where $U : \mathbb{R}^d \to \mathbb{R}$ is called the potential energy function. In practice, directly generating samples from $\pi$ turns out to be intractable except for very few special cases, therefore one often needs to resort to approximate methods. A popular way to approximately generate samples from $\pi$ is based on discretizations of a stochastic differential equation (SDE) that has $\pi$ as an invariant distribution [14]. A common choice is the over-damped Langevin equation associated with $\pi$, that is the stochastic differential equation (SDE) given by

$$\mathrm{d}\vartheta_t = -\nabla U(\vartheta_t)\mathrm{d}t + \sqrt{2}\mathrm{d}B_t \;, \tag{1}$$

where $(B_t)_{t\geq 0}$ is the standard $d$-dimensional Brownian motion. Under mild assumptions on $U$ (cf. [2]), $(\vartheta_t)_{t\geq 0}$ is a well defined Markov process which is geometrically ergodic with respect to $\pi$. Therefore, if continuous sample paths from $(\vartheta_t)_{t\geq 0}$ could be generated, they could be used as approximate samples from $\pi$. However, this is not possible and therefore in practice we need to use a discretization of (1). The most common discretization is the Euler-Maruyama scheme, which boils down to applying the following update equation iteratively: $\theta_{k+1} = \theta_k - \gamma_{k+1}\nabla U(\theta_k) + \sqrt{2\gamma_{k+1}}Z_{k+1}$, for $k \geq 0$ with initial state $\theta_0$. Here, $(\gamma_k)_{k\geq 1}$ is a sequence of non-increasing step sizes and $(Z_k)_{k\geq 1}$ is a sequence of independent and identically distributed (i.i.d.) $d$-dimensional standard normal random variables. This schema is called the Unadjusted Langevin Algorithm (ULA) [2]. When the sequence of the step sizes $(\gamma_k)_{k\geq 0}$ goes to 0 as $k$ goes to infinity, it has been shown in [15] and [16] that the empirical distribution of $(\theta_k)_{k\geq 0}$ weakly converges to $\pi$ under certain assumptions. A central limit theorem for additive functionals has also been obtained in [17] and [16].

In Bayesian machine learning, $\pi$ is often chosen as the Bayesian posterior, which imposes the following form on the potential energy: $U(\theta) = -(\sum_{n=1}^{N} \log \mathsf{p}(\mathbf{x}_n|\theta) + \log \mathsf{p}(\theta))$ for all $\theta \in \mathbb{R}^d$, where $\mathbf{x} \equiv \{\mathbf{x}_n\}_{n=1}^{N}$ is a set of observed i.i.d. data points, belonging to $\mathbb{R}^m$, for $m \geq 1$, $\mathsf{p}(\mathbf{x}_n|\cdot) : \mathbb{R}^d \to \mathbb{R}_+^*$ is the likelihood function, and $\mathsf{p}(\theta) : \mathbb{R}^d \to \mathbb{R}_+^*$ is the prior distribution. In large scale settings, $N$ becomes very large and therefore computing $\nabla U$ can be computationally very demanding, limiting the applicability of ULA. Inspired by stochastic optimization techniques, in [1], the authors have proposed replacing the exact gradient $\nabla U$ with an unbiased estimator and presented the SGLD algorithm that iteratively applies the following update equation:

$$\theta_{k+1} = \theta_k - \gamma_{k+1}\nabla \tilde{U}_{k+1}(\theta_k) + \sqrt{2\gamma_{k+1}}Z_{k+1} \;, \tag{2}$$

where $(\nabla \tilde{U}_k)_{k \geq 1}$ is a sequence of i.i.d. unbiased estimators of $\nabla U$. In the following, the common distribution of $(\nabla \tilde{U}_k)_{k \geq 1}$ will be denoted by $\mathcal{L}$. A typical choice for the sequence of estimators $(\nabla \tilde{U}_k)_{k \geq 1}$ of $\nabla U$ is to randomly draw an i.i.d. sequence of data subsample $(\mathbf{R}_k)_{k \geq 1}$ with $\mathbf{R}_k \subset [N] = \{1, \ldots, N\}$ having a fixed number of elements $|\mathbf{R}_k| = B$ for all $k \geq 1$. Then, set for all $\theta \in \mathbb{R}^d, k \geq 1$

$$\nabla \tilde{U}_k(\theta) = -[\nabla \log \mathsf{p}(\theta) + \frac{N}{B} \sum\nolimits_{i \in \mathbf{R}_k} \nabla \log \mathsf{p}(\mathbf{x}_i|\theta)] . \tag{3}$$

Convergence analysis of SGLD has been studied in [18, 19] and it has been shown in [20] that for constant step sizes $\gamma_k = \gamma > 0$ for all $k \geq 1$, the bias and the MSE of SGLD are of order $\mathcal{O}(\gamma + 1/(\gamma K))$ and $\mathcal{O}(\gamma^2 + 1/(\gamma K))$, respectively. Recently, it has been shown that these bounds are also valid in a more general family of SG-MCMC methods [10].

## 2.2 Richardson-Romberg Extrapolation for SDEs

Richardson-Romberg extrapolation is a well-known method in numerical analysis, which aims to improve the rate of convergence of a sequence. Talay and Tubaro [21] showed that the rate of convergence of Monte Carlo estimates on certain SDEs can be radically improved by using an RR extrapolation that can be described as follows. Let us consider the SDE in (1) and its Euler discretization with exact gradients and fixed step size, *i.e.* $\gamma_k = \gamma > 0$ for all $k \geq 1$. Under mild assumptions on $U$ (cf. [22]), the homogeneous Markov chain $(\theta_k)_{k \geq 0}$ is ergodic with a unique invariant distribution $\pi_\gamma$, which is different from the target distribution $\pi$. However, [21] showed that for $f$ sufficiently smooth with polynomial growth, there exists a constant $C$, which only depends on $\pi$ and $f$ such that $\pi_\gamma(f) = \pi(f) + C\gamma + \mathcal{O}(\gamma^2)$, where $\pi(f) = \int_{\mathbb{R}^d} f(x)\pi(\mathrm{d}x)$. By exploiting this result, RR extrapolation suggests considering two different discretizations of the same SDE with two different step sizes $\gamma$ and $\gamma/2$. Then instead of $\pi_\gamma(f)$, if we consider $2\pi_{\gamma/2}(f) - \pi_\gamma(f)$ as the estimator, we obtain $\pi(f) - (2\pi_{\gamma/2}(f) - \pi_\gamma(f)) = \mathcal{O}(\gamma^2)$. In the case where the sequence $(\gamma_k)_{k \geq 0}$ goes to 0 as $k \to +\infty$, it has been observed in [23] that the estimator defined by RR extrapolation satisfies a CLT. The applications of RR extrapolation to SG-MCMC have not yet been explored.

# 3 Stochastic Gradient Richardson-Romberg Langevin Dynamics

In this study, we explore the use of RR extrapolation in SG-MCMC algorithms for improving their rates of convergence. In particular, we focus on the applications of RR extrapolation on the SGLD estimator and present a novel SG-MCMC algorithm referred to as Stochastic Gradient Richardson-Romberg Langevin Dynamics (SGRRLD).

The proposed algorithm applies RR extrapolation on SGLD by considering two SGLD chains applied to the SDE (1), with two different sequences of step sizes satisfying the following relation. For the first chain, we consider a sequence of non-increasing step sizes $(\gamma_k)_{k \geq 1}$ and for the second chain, we use the sequence of step sizes $(\eta_k)_{k \geq 1}$ defined by $\eta_{2k-1} = \eta_{2k} = \gamma_k/2$ for $k \geq 1$. These two chains are started at the same point $\theta_0 \in \mathbb{R}^d$, and are run accordingly to (2) but the chain with the smallest step size is run twice more time than the other one. In other words, these two discretizations are run until the same time horizon $\sum_{k=1}^K \gamma_k$, where $K$ is the number of iterations. Finally, we extrapolate the two SGLD estimators in order to construct the new one. Each iteration of SGRRLD will consist of one step of the first SGLD chain with $(\gamma_k)_{k \geq 1}$ and two steps of the second SGLD chain with $(\eta_k)_{k \geq 1}$. More formally the proposed algorithm is defined by: consider a starting point $\theta_0^{(\gamma)} = \theta_0^{(\gamma/2)} = \theta_0$ and for $k \geq 0$,

$$\text{Chain 1:} \quad \theta_{k+1}^{(\gamma)} = \theta_k^{(\gamma)} - \gamma_{k+1} \nabla \tilde{U}_{k+1}^{(\gamma)}(\theta_k^{(\gamma)}) + \sqrt{2\gamma_{k+1}} Z_{k+1}^{(\gamma)} , \tag{4}$$

$$\text{Chain 2:} \quad \begin{cases} \theta_{2k+1}^{(\gamma/2)} = \theta_{2k}^{(\gamma/2)} - \frac{\gamma_{k+1}}{2} \nabla \tilde{U}_{2k+1}^{(\gamma/2)}(\theta_{2k+1}^{(\gamma/2)}) + \sqrt{\gamma_{k+1}} Z_{2k+1}^{(\gamma/2)} \\ \theta_{2k+2}^{(\gamma/2)} = \theta_{2k+1}^{(\gamma/2)} - \frac{\gamma_{k+1}}{2} \nabla \tilde{U}_{2k+2}^{(\gamma/2)}(\theta_{2k+1}^{(\gamma/2)}) + \sqrt{\gamma_{k+1}} Z_{2k+2}^{(\gamma/2)} \end{cases} \tag{5}$$

where $(Z_k^{(\gamma/2)})_{k \geq 1}$ and $(Z_k^{(\gamma)})_{k \geq 1}$ are two sequences of $d$-dimensional i.i.d. standard Gaussian random variables and $(\nabla \tilde{U}_k^{(\gamma/2)})_{k \geq 1}, (\nabla \tilde{U}_k^{(\gamma)})_{k \geq 1}$ are two sequences of i.i.d. unbiased estimators of $\nabla U$ with the same common distribution $\mathcal{L}$, meaning that the mini-batch size has to be the same.

For a test function $f : \mathbb{R}^d \to \mathbb{R}$, we then define the estimator of $\pi(f)$ based on RR extrapolation as follows: (for all $K \in \mathbb{N}^*$)

$$\hat{\pi}_K^{\mathrm{R}}(f) = \left( \sum_{k=2}^{K+1} \gamma_k \right)^{-1} \sum_{k=1}^{K} \gamma_{k+1} \left[ \{ f(\theta_{2k-1}^{(\gamma/2)}) + f(\theta_{2k}^{(\gamma/2)}) \} - f(\theta_k^{(\gamma)}) \right] , \tag{6}$$

We provide a pseudo-code of SGRRLD in the supplementary document.

Under mild assumptions on $\nabla U$ and the law $\mathcal{L}$ (see the conditions in the Supplement), by [19, Theorem 7] we can show that $\hat{\pi}_K^{\mathrm{R}}(f)$ is a consistent estimator of $\pi(f)$: when $\lim_{k \to +\infty} \gamma_k = 0$ and $\lim_{K \to +\infty} \sum_{k=1}^{K} \gamma_{k+1} = +\infty$, then $\lim_{K \to +\infty} \hat{\pi}_K^{\mathrm{R}}(f) = \pi(f)$ almost surely. However, it is not immediately clear whether applying an RR extrapolation would provide any advantage over SGLD in terms of the rate of convergence. Even if RR extrapolation were to reduce the bias of the SGLD estimator, this improvement could be offset by an increase of variace. In the context of a general class of SDEs, in [13] it has been shown that the variance of estimator based on RR extrapolation can be controlled by using correlated Brownian increments and the best choice in this sense is in fact taking the two sequences $(Z_k^{(\gamma/2)})_{k \geq 1}$ and $(Z_k^{(\gamma)})_{k \geq 1}$ perfectly correlated, *i.e.* for all $k \geq 1$,

$$Z_k^{(\gamma)} = (Z_{2k-1}^{(\gamma/2)} + Z_{2k}^{(\gamma/2)})/\sqrt{2} . \tag{7}$$

This choice has also been justified in the context of the sampling of the stationary distribution of a diffusion in [23] through a central limit theorem.

Inspired by [23], in order to be able to control the variance of the SGRRLD estimator, we consider correlated Brownian increments. In particular, we assume that the Brownian increments in (4) and (5) satisfy the following relationship: there exist a matrix $\Sigma \in \mathbb{R}^{d \times d}$, a sequence $(W_k)_{k \geq 1}$ of $d$ dimensional i.i.d. standard Gaussian random variables, independent of $(Z_k^{(\gamma/2)})_{k \geq 1}$ such that $\mathrm{Id} - \Sigma^\top \Sigma$ is a positive semidefinite matrix and for all $k \geq 0$,

$$Z_{k+1}^{(\gamma)} = \Sigma^\top (Z_{2k+1}^{(\gamma/2)} + Z_{2(k+1)}^{(\gamma/2)})/\sqrt{2} + (\mathrm{Id} - \Sigma^\top \Sigma)^{1/2} W_{k+1} , \tag{8}$$

where Id denotes the identity matrix. In Section 4, we will show that the properly scaled SGRRLD estimator converges to a Gaussian random variable whose variance is minimal when $\Sigma = \mathrm{Id}$, and therefore $Z_{k+1}^{(\gamma)}$ should be chosen as in (7). Accordingly, (8) justifies the choice of using the same Brownian motion in the two discretizations, extending the results of [23] to SG-MCMC. On the other hand, regarding the sequences of estimators for $\nabla U$, we assume that they can also be correlated but do not assume an explicit form on their relation. However, it is important to note that if the two sequences $(\nabla \tilde{U}_k^{(\gamma/2)})_{k \geq 1}$ and $(\nabla \tilde{U}_k^{(\gamma)})_{k \geq 1}$ do not have the same common distribution, then the SGRRLD estimator can have a bias, which would have the same order as of vanilla SGLD (with the same sequence of step sizes). In the particular case of (3), in order for SGRRLD to gain efficiency compared to SGLD, the mini-batch size has to be the same for the two chains.

## 4  Convergence Analysis

We analyze asymptotic and non-asymptotic properties of SGRRLD. In order to save space and avoid obscuring the results, we present the technical conditions under which the theorems hold, and the full proofs in the supplementary document.

We first present a central limit theorem for the estimator $\hat{\pi}_K^{\mathrm{R}}(f)$ of $\pi(f)$ (see (6)) for a smooth function $f$. Let us define $\Gamma_K^{(n)} = \sum_{k=1}^{K} \gamma_{k+1}^n$ and $\Gamma_K = \Gamma_K^{(1)}$, for all $n \in \mathbb{N}$.

**Theorem 1.** *Let $f : \mathbb{R}^d \to \mathbb{R}$ be a smooth function and $(\gamma_k)_{k \geq 1}$ be a nonincreasing sequence satisfying $\lim_{k \to +\infty} \gamma_k = 0$ and $\lim_{K \to +\infty} \Gamma_K = +\infty$. Let $(\theta_k^{(\gamma)}, \theta_k^{(\gamma/2)})_{k \geq 0}$ be defined by (4)-(5), started at $\theta_0 \in \mathbb{R}^d$ and assume that the relation (8) holds for $\Sigma \in \mathbb{R}^{d \times d}$. Under appropriate conditions on $U$, $f$ and $\mathcal{L}$, then the following statements hold:*

*a) If $\lim_{K \to +\infty} \Gamma_K^{(3)}/\sqrt{\Gamma_K} = 0$, then $\sqrt{\Gamma_K}(\hat{\pi}_K^{\mathrm{R}}(f) - \pi(f))$ converges in law as $K$ goes to infinity to a zero-mean Gaussian random variable with variance $\sigma_{\mathrm{R}}^2$, which is minimized when $\Sigma = \mathrm{Id}$.*

*b) If $\lim_{K \to +\infty} \Gamma_K^{(3)}/\sqrt{\Gamma_K} = \kappa \in (0, +\infty)$, then $\sqrt{\Gamma_K}(\hat{\pi}_K^{\mathrm{R}}(f) - \pi(f))$ converges in law as $K$ goes to infinity to a Gaussian random variable with variance $\sigma_{\mathrm{R}}^2$ and mean $\kappa \mu_{\mathrm{R}}$.*

*c) If* $\lim_{K \to +\infty} \Gamma_K^{(3)}/\sqrt{\Gamma_K} = +\infty$, *then* $(\Gamma_K/\Gamma_k^{(3)})(\hat{\pi}_K^{\mathrm{R}}(f) - \pi(f))$ *converges in probability as K goes to infinity to* $\mu_{\mathrm{R}}$.

*The expressions of* $\sigma_{\mathrm{R}}^2$ *and* $\mu_{\mathrm{R}}$ *are given in the supplementary document.*

*Proof (Sketch).* The proof follows the same strategy as the one in [23, Theorem 4.3] for ULA. We assume that the Poisson equation associated with $f$ has a solution $g \in C^9(\mathbb{R}^d)$. Then, the proof consists in making a 7th order Taylor expansion for $g(\theta_{k+1}^{(\gamma)})$, $g(\theta_{2k}^{(\gamma/2)})$ and $g(\theta_{2k+1}^{(\gamma)})$ at $\theta_k^{(\gamma)}$, $\theta_{2k-1}^{(\gamma/2)}$ and $\theta_{2k}^{(\gamma/2)}$, respectively. Then $\hat{\pi}_K^{\mathrm{R}}(f) - \pi(f)$ is decomposed as a sum of three terms $A_{1,K} + A_{2,K} + A_{3,K}$. $A_{1,K}$ is the fluctuation term and $\Gamma_K^{1/2} A_{1,K}$ converges to a zero-mean Gaussian random variable with variance $\sigma_{\mathrm{R}}^2$. $A_{2,K}$ is the bias term, and $\Gamma_K A_{2,K}/\Gamma_K^{(3)}$ converges in probability to $\mu_{\mathrm{R}}$ as $K$ goes to $+\infty$ if $\lim_{K \to +\infty} \Gamma_K^{(3)} = +\infty$. Finally the last term $\Gamma_K^{1/2} A_{3,K}$ goes to 0 as $K$ goes to $+\infty$. The detailed proof is given in the supplementary document. $\square$

These results state that the Gaussian noise dominates the stochastic gradient noise. Moreover, we also observe that the correlation between the two sequences of Gaussian random variables $(Z_k^{(\gamma)})_{k \geq 1}$ and $(Z_k^{(\gamma/2)})_{k \geq 1}$ has an important impact on the asymptotic convergence of $\hat{\pi}^{\mathrm{R}}(f)$, whereas the correlation of the two sequences of stochastic gradients does not.

A typical choice of decreasing sequence $(\gamma_k)_{k \geq 1}$ is of the form $\gamma_k = \gamma_1 k^{-\alpha}$ for $\alpha \in (0, 1]$. With such a choice, Theorem 1 states that $\hat{\pi}^{\mathrm{R}}(f)$ converges to $\pi(f)$ at a rate of convergence of order $\mathcal{O}(K^{-((1-\alpha)/2) \wedge (2\alpha)})$, where $a \wedge b = \min(a, b)$. Therefore, the optimal choice for the exponent $\alpha$ for obtaining the fastest convergence turns out to be $\alpha = 1/5$, which implies a rate of convergence of order $\mathcal{O}(K^{-2/5})$. Note that this rate is higher than SGLD whose optimal rate is of order $\mathcal{O}(K^{-1/3})$. Besides, $\alpha = 1/5$ corresponds to the second point of Theorem 1, in which there is an equal contribution of the bias and the fluctuation at an asymptotic level. Futher discussions and detailed calculations can be found in the supplementary document.

We now derive non-asymptotic bounds for the bias and the MSE of the estimator $\hat{\pi}^{\mathrm{R}}(f)$.

**Theorem 2.** *Let* $f : \mathbb{R}^d \to \mathbb{R}$ *be a smooth function and* $(\gamma_k)_{k \geq 1}$ *be a nonincreasing sequence such that there exists* $K_1 \geq 1$, $\gamma_{K_1} \leq 1$ *and* $\lim_{K \to +\infty} \Gamma_K = +\infty$. *Let* $(\theta_k^{(\gamma)}, \theta_k^{(\gamma/2)})_{k \geq 0}$ *be defined by* (4)-(5), *started at* $\theta_0 \in \mathbb{R}^d$. *Under appropriate conditions on U, f and* $\mathcal{L}$, *then there exists* $C \geq 0$ *such that for all* $K \in \mathbb{N}$, $K \geq 1$:

$$\text{BIAS:} \quad \left| \mathbb{E}\left[ \hat{\pi}_K^{\mathrm{R}}(f) - \pi(f) \right] \right| \leq (C/\Gamma_K) \left\{ \Gamma_K^{(3)} + 1 \right\}$$

$$\text{MSE:} \quad \mathbb{E}\left[ \left\{ \hat{\pi}_K^{\mathrm{R}}(f) - \pi(f) \right\}^2 \right] \leq C \{ (\Gamma_K^{(3)}/\Gamma_K)^2 + 1/\Gamma_K \}.$$

*Proof (Sketch).* The proof follows the same strategy as the one of Theorem 1, but instead of establishing the exact convergence of the fluctuation and the bias terms, we just give an upper bound for these two terms. The detailed proof is given in the supplementary document. $\square$

It is important to observe that the constant $C$ which appears in Theorem 2 depends on moments of the estimator of the gradient. For fixed step size $\gamma_k = \gamma$ for all $k \geq 1$, Theorem 2 shows that the bias is of order $\mathcal{O}(\gamma^2 + 1/(K\gamma))$. Therefore, if the number of iterations $K$ is fixed then the choice of $\gamma$ which minimizes this bound is $\gamma \propto K^{-1/3}$, obtained by differentiating $x \mapsto x^2 + (xK)^{-1}$. Choosing this value for $\gamma$ leads to the optimal rate for the bias of order $\mathcal{O}(K^{-2/3})$. Note that this bound is better than SGLD for which the optimal bound of the bias at fixed $K$ is of order $\mathcal{O}(K^{-1/2})$. The same approach can be applied to the MSE which is of order $\mathcal{O}(\gamma^4 + 1/(K\gamma))$. Then, the optimal choice of the step size is $\gamma = \mathcal{O}(K^{-1/5})$, leading to a bound of order $\mathcal{O}(K^{-4/5})$. Similar to the previous case, this bound is smaller than the bound obtained with SGLD, which is $\mathcal{O}(K^{-2/3})$.

If we choose $\gamma_k = \gamma_1 k^{-\alpha}$ for $\alpha \in (0, 1]$, Theorem 2 shows that the bias and the MSE go to 0 as $K$ goes to infinity. More precisely for $\alpha \in (0, 1)$, the bound for the bias is $\mathcal{O}(K^{-(2\alpha) \wedge (1-\alpha)})$, and is therefore minimal for $\alpha = 1/3$. As for the MSE, the bound provided by Theorem 2 is $\mathcal{O}(K^{-(4\alpha) \wedge (1-\alpha)})$ which is consistent with Theorem 1, leading to an optimal bound of order $\mathcal{O}(K^{-4/5})$ as $\alpha = 1/5$.

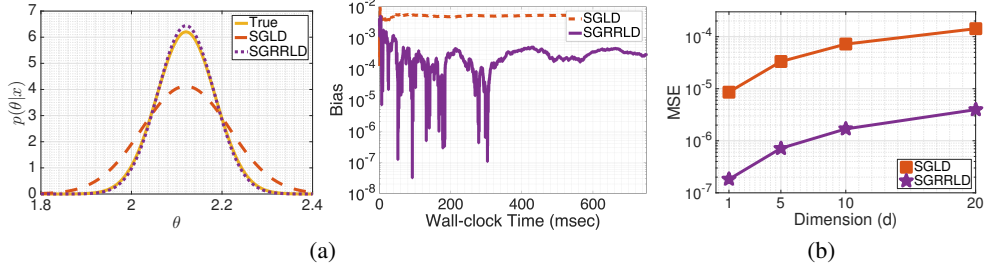

Figure 1: The performance of SGRRLD on synthetic data. (a) The true posterior and the estimated posteriors. (b) The MSE for different problem sizes.

## 5 Experiments

### 5.1 Linear Gaussian Model

We conduct our first set of experiments on synthetic data where we consider a simple Gaussian model whose posterior distribution is analytically available. The model is given as follows:

$$\theta \sim \mathcal{N}(0, \sigma_\theta^2 \, \mathrm{Id}) \ , \quad \mathbf{x}_n | \theta \sim \mathcal{N}(\mathbf{a}_n^\top \theta, \sigma_\mathbf{x}^2) \ , \text{ for all } n \ . \tag{9}$$

Here, we assume that the explanatory variables $\{\mathbf{a}_n\}_{n=1}^N \in \mathbb{R}^{N \times d}$, $\sigma_\theta^2$ and $\sigma_\mathbf{x}^2$ are known and we aim to draw samples from the posterior distribution $p(\theta|\mathbf{x})$. In all the experiments, we first randomly generate $\mathbf{a}_n \sim \mathcal{N}(0, 0.5\,\mathrm{Id})$ and we generate the true $\theta$ and the response variables $\mathbf{x}$ by using the generative model given in (9). All our experiments are conducted on a standard laptop computer with 2.5GHz Quad-core Intel Core i7 CPU, and in all settings, the two chains of SGRRLD are run in parallel.

In our first experiment, we set $d = 1$, $\sigma_\theta^2 = 10$, $\sigma_\mathbf{x}^2 = 1$, $N = 1000$, and the size of each minibatch $B = N/10$. We fix the step size to $\gamma = 10^{-3}$. In order to ensure that both algorithms are run for a fixed computation time, we run SGLD for $K = 21000$ iterations where we discard the first 1000 samples as burn-in, and we run SGRRLD for $K = 10500$ iterations accordingly, where we discard the samples generated in the first 500 iterations as burn-in. Figure 1(a) shows the typical results of this experiment. In particular, in the left figure, we illustrate the true posterior distribution and the Gaussian density $\mathcal{N}(\hat{\mu}_{\mathrm{post}}, \hat{\sigma}_{\mathrm{post}}^2)$ for both algorithms, where $\hat{\mu}_{\mathrm{post}}$ and $\hat{\sigma}_{\mathrm{post}}^2$ denote the empirical posterior mean and variance, respectively. In the right figure, we monitor the bias of the estimated variance as a function of computation time. The results show that SGLD overestimates the posterior variance, whereas SGRRLD is able to reduce this error significantly. We also observe that the results support our theory: the bias of the estimated variance is $\approx 10^{-2}$ for SGLD whereas this bias is reduced to $\approx 10^{-4}$ with SGRRLD.

In our second experiment, we fix $\gamma$ and $K$ and monitor the MSE of the posterior covariance as a function of the dimension $d$ of the problem. In order to measure the MSE, we compute the squared Frobenius norm of the difference between the true posterior covariance and the estimated covariance. Similarly to the previous experiment, we average 100 runs that are initialized randomly. The results are shown in Figure 1(b). The re-

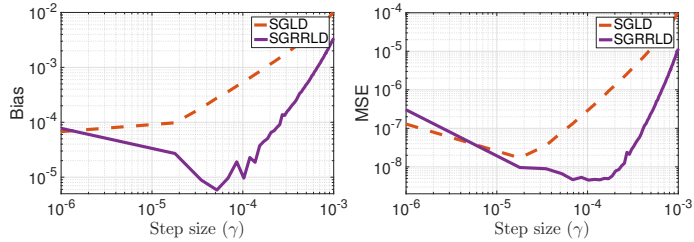

Figure 2: Bias and MSE of SGLD and SGRRLD for different step sizes.

sults clearly show that SGRRLD provides significant performance improvement over SGLD, where the MSE of SGRRLD is in the order of the square of the MSE of SGLD for all values of $d$.

In our next experiment, we use the same setting as in the first experiment and we monitor the bias and the MSE of the estimated variance as a function of the step size $\gamma$. For evaluation, we average 100 runs that are initialized randomly. As depicted in Figure 2, the results show that SGRRLD yields

significantly better results than SGLD in terms of both the bias and MSE. Note that for very small $\gamma$, the bias and MSE increase. This is due to the term $1/(K\gamma)$ in the bounds of Theorem 2 dominates both the bias and the MSE as expected since $K$ is fixed. Therefore, we observe a drop in the bias and the MSE as we increase $\gamma$ up to $\approx 8 \times 10^{-5}$, and then they gradually increase along with $\gamma$.

We conduct the next experiment in order to check the rate of convergence that we have derived in Theorem 2 for fixed step size $\gamma_k = \gamma$ for all $k \geq 1$. We observe that the optimal choice for the step size is of the form $\gamma = \gamma_b^\star K^{-1/3}$ and $\gamma = \gamma_M^\star K^{-0.2}$ for the bias and MSE, respectively. To confirm our findings, we first need to determine the constants $\gamma_b^\star$ and $\gamma_M^\star$, which can be done by using the results from the previous experiment. Accordingly, we observe that $\gamma_b^\star \approx 8.5 \cdot 10^{-5} \cdot$

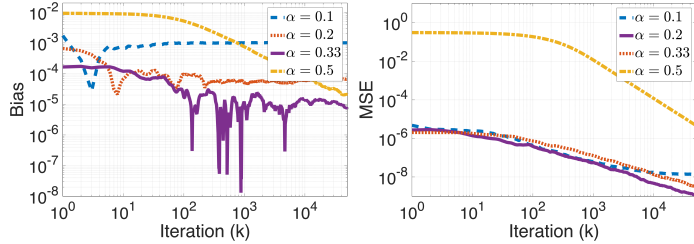

Figure 3: Bias and MSE of SGRRLD with different rates for step size ($\alpha$).

$(20000)^{1/3} \approx 2 \cdot 10^{-3}$ and $\gamma_M^\star \approx 1.7 \cdot 10^{-4} \cdot (20000)^{0.2} \approx 10^{-3}$. Then, to confirm the right dependency of $\gamma$ on $K$, we fix $K = 10^6$ and monitor the bias with the sequence of step sizes $\gamma = \gamma_b^\star K^{-\alpha}$ and the MSE with $\gamma = \gamma_M K^{-\alpha}$ for several values of $\alpha$ as given in Figure 3. It can be observed that the optimal convergence rate is still obtained for $\alpha = 1/3$ for the bias and $\alpha = 0.2$ for the MSE, which confirms the results of Theorem 2. For a decreasing sequence of step sizes $\gamma_k = \gamma_1^\star k^\alpha$ for $\alpha \in (0, 1]$, we conduct a similar experiment to confirm that the best convergence rate is achieved choosing $\alpha = 1/3$ in the case of the bias and $\alpha = 0.2$ in the case of the MSE. The resulting figures can be found in the supplementary document.

In our last synthetic data experiment, instead of SGLD, we consider another SG-MCMC algorithm, namely the Stochastic Gradient Hamiltonian Monte Carlo (SGHMC) [6]. We apply the proposed extrapolation scheme described in Section 3 to SGHMC and call the resulting algorithm Stochastic Gradient Richardson-Romberg Hamiltonian Monte Carlo (SGRRHMC). In this experiment, we use the same setting as we use in Figure 2, and

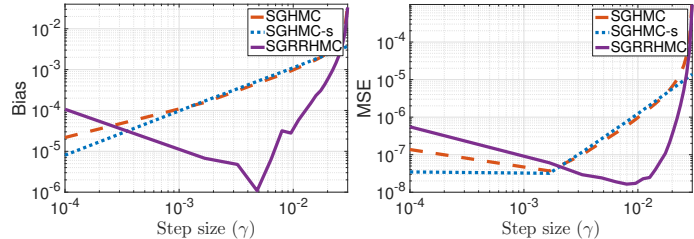

Figure 4: The performance of RR extrapolation on SGHMC.

we monitor the bias and the MSE of the estimated variance as a function of $\gamma$. We compare SGR-RHMC against SGHMC with Euler discretization [6] and SGHMC with an higher-order splitting integrator (SGHMC-s) [10] (we describe SGHMC, SGHMC-s, and SGRRHMC in more detail in the supplementary document). We average 100 runs that are initialized randomly. As given in Figure 4, the results are similar to the ones obtained in Figure 2: for large enough $\gamma$, SGRRHMC yields significantly better results than SGHMC. For small $\gamma$, the term $1/(K\gamma)$ in the bound derived in Theorem 2 dominates the MSE and therefore SGRRHMC requires a larger $K$ for improving over SGHMC. For large enough values of $\gamma$, we observe that SGRRHMC obtains an MSE similar to that of SGHMC-s with small $\gamma$, which confirms our claim that the proposed approach can achieve the accuracy of the methods that are based on higher-order integrators.

## 5.2 Large-Scale Matrix Factorization

In our second set of experiments, we evaluate our approach on a large-scale matrix factorization problem for a link prediction application, where we consider the following probabilistic model: $W_{ip} \sim \mathcal{N}(0, \sigma_w^2)$, $H_{pj} \sim \mathcal{N}(0, \sigma_h^2)$, $X_{ij}|W, H \sim \mathcal{N}\left(\sum_p W_{ip}H_{pj}, \sigma_x^2\right)$, where $X \in \mathbb{R}^{I \times J}$ is the observed data matrix with missing entries, and $W \in \mathbb{R}^{I \times P}$ and $H \in \mathbb{R}^{D \times P}$ are the latent factors,

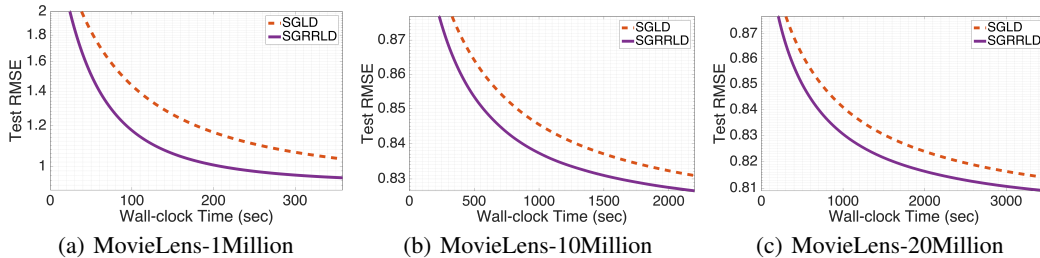

(a) MovieLens-1Million      (b) MovieLens-10Million      (c) MovieLens-20Million

Figure 5: The performance of SGRRLD on large-scale matrix factorization problems.

whose entries are i.i.d. distributed. The aim in this application is to predict the missing values of $X$ by using a low-rank approximation. This model is similar to the Bayesian probabilistic matrix factorization model [24] and it is often used in large-scale matrix factorization problems [25], in which SG-MCMC has been shown to outperform optimization methods such as SGD [26].

In this experiment, we compare SGRRLD against SGLD on three large movie ratings datasets, namely the MovieLens 1Million (ML-1M), MovieLens 10Million (ML-10M), and MovieLens 20Million (ML-20M) (`grouplens.org`). The ML-1M dataset contains about $1$ million ratings applied to $I = 3883$ movies by $J = 6040$ users, resulting in a sparse observed matrix $X$ with $4.3\%$ non-zero entries. The ML-10M dataset contains about $10$ million ratings applied to $I = 10681$ movies by $J = 71567$ users, resulting in a sparse observed matrix $X$ with $1.3\%$ non-zero entries. Finally, The ML-20M dataset contains about $20$ million ratings applied to $I = 27278$ movies by $J = 138493$ users, resulting in a sparse observed matrix $X$ with $0.5\%$ non-zero entries. We randomly select $10\%$ of the data as the test set and use the remaining data for generating the samples. The rank of the factorization is chosen as $P = 10$. We set $\sigma_w^2 = \sigma_h^2 = \sigma_x^2 = 1$. For all datasets, we use a constant step size. We run SGLD for $K = 10500$ iterations where we discard the first $500$ samples as burn-in. In order to keep the computation time the same, we run SGRRLD for $K = 5250$ iterations where we discard the first $250$ iterations as burn-in. For ML-1M we set $\gamma = 2 \times 10^{-6}$ and for ML-10M and ML-20M we set $\gamma = 2 \times 10^{-5}$. The size of the subsamples $B$ is selected as $N/10$, $N/50$, and $N/500$ for ML-1M, ML-10M and ML-20M, respectively. We have implemented SGLD and SGRRLD in C by using the GNU Scientific Library for efficient matrix computations. We fully exploit the inherently parallel structure of SGRRLD by running the two chains in parallel as two independent processes, whereas SGLD cannot benefit from this parallel computation architecture due to its inherently sequential nature. Therefore their wall-clock times are nearly exactly the same.

Figure 5 shows the comparison of SGLD and SGRRLD in terms of the root mean squared-errors (RMSE) that are obtained on the test sets as a function of wall-clock time. The results clearly show that in all datasets SGRRLD yields significant performance improvements. We observe that in the ML-1M experiment SGRRLD requires only $\approx 200$ seconds for achieving the accuracy that SGLD provides after $\approx 400$ seconds. We see similar behaviors in the ML-10M and ML-20M experiments: SGRRLD appears to be more efficient than SGLD. The results indicate that by using our approach, we either obtain the same accuracy of SGLD in shorter time or we obtain a better accuracy by spending the same amount of time as SGLD.

## 6 Conclusion

We presented SGRRLD, a novel scalable sampling algorithm that aims to reduce the bias of SG-MCMC while keeping the variance at a reasonable level by using RR extrapolation. We provided formal theoretical analysis and showed that SGRRLD is asymptotically consistent and satisfies a central limit theorem. We further derived bounds for its non-asymptotic bias and the mean squared-error, and showed that SGRRLD attains higher rates of convergence than all known SG-MCMC methods with first-order integrators in both finite-time and asymptotically. We supported our findings using both synthetic and real data experiments, where SGRRLD appeared to be more efficient than SGLD in terms of computation time on a large-scale matrix factorization application. As a next step, we plan to explore the use of the multi-level Monte Carlo approaches [27] in our framework.

**Acknowledgements:** This work is partly supported by the French National Research Agency (ANR) as a part of the EDISON 3D project (ANR-13-CORD-0008-02).

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
