[Supplementary Material]

# Stochastic Gradient Richardson-Romberg Markov Chain Monte Carlo

**SUPPLEMENTARY DOCUMENT**

**Alain Durmus[1], Umut Şimşekli[1], Éric Moulines[2], Roland Badeau[1], Gaël Richard[1]**
1: LTCI, CNRS, Télécom ParisTech, Université Paris-Saclay, 75013, Paris, France
2: Centre de Mathématiques Appliquées, UMR 7641, École Polytechnique, France

## 1 Pseudocode of Stochastic Gradient Richardson-Romberg Langevin Dynamics

In this section, we provide a pseudocode of SGRRLD with fixed step size in Algorithm 1 and a pseudocode of SGRRLD with decreasing step size in Algorithm 2.

---

**Algorithm 1:** Stochastic Gradient Richardson-Romberg Langevin Dynamics with fixed step size.

---

**Input** : Random number seed $\mathcal{S}$, Step size $\gamma$, Initial state $\theta_0^{(\gamma/2)} = \theta_0^{(\gamma)} = \theta_0$, Test function $f(\cdot)$,
Number of iterations $K$, a probability distribution $\mathcal{L}$ of an unbiased estimators $\nabla\tilde{U}$ for $\nabla U$

**Output** : $\hat{\pi}_K^{\mathrm{R}}(f) \approx \int f(\theta)\pi(d\theta)$

```
// Run two chains in parallel with consistent Brownian increments
```

**Chain 1:**

**for** $k = 1, \cdots, K$ **do**
   Set random number generator seed to $\mathcal{S}$
   Draw $Z_{2k-1}^{(\gamma/2)} \sim \mathcal{N}(0, \mathrm{Id})$, $Z_{2k}^{(\gamma/2)} \sim \mathcal{N}(0, \mathrm{Id})$
   Set $Z_k^{(\gamma)} = (Z_{2k-1}^{(\gamma/2)} + Z_{2k}^{(\gamma/2)})/\sqrt{2}$
   Draw $\nabla\tilde{U}_k^{(\gamma)}$ from $\mathcal{L}$
   $\theta_k^{(\gamma)} = \theta_{k-1}^{(\gamma)} - \gamma\nabla\tilde{U}_k^{(\gamma)}(\theta_{k-1}^{(\gamma)}) + \sqrt{2\gamma}Z_k^{(\gamma)}$

Compute $\hat{\pi}_K^{(\gamma)}(f) = \frac{1}{K}\sum_{k=1}^{K} f(\theta_k^{(\gamma)})$

**Chain 2:**

**for** $k = 1, \cdots, 2K$ **do**
   Set random number generator seed to $\mathcal{S}$
   Draw $Z_k^{(\gamma/2)} \sim \mathcal{N}(0, \mathrm{Id})$
   Draw $\nabla\tilde{U}_k^{(\gamma/2)}$ from $\mathcal{L}$
   $\theta_k^{(\gamma/2)} = \theta_{k-1}^{(\gamma/2)} - \frac{\gamma}{2}\nabla\tilde{U}_k^{(\gamma/2)}(\theta_{k-1}^{(\gamma/2)})$
           $+ \sqrt{\gamma}Z_k^{(\gamma/2)}$

Compute $\hat{\pi}_K^{(\gamma/2)}(f) = \frac{1}{2K}\sum_{k=1}^{2K} f(\theta_k^{(\gamma/2)})$

```
// RR extrapolation.
```

$\hat{\pi}_K^{\mathrm{R}}(f) = 2\hat{\pi}_K^{(\gamma/2)}(f) - \hat{\pi}_K^{(\gamma)}(f)$

---

**Algorithm 2:** Stochastic Gradient Richardson-Romberg Langevin Dynamics with a sequence of step sizes.

---

**Input** : Random number seed $\mathcal{S}$, Sequence of step sizes $(\gamma_k)_{k\geq 1}$, Initial state $\theta_0^{(\gamma/2)} = \theta_0^{(\gamma)} = \theta_0$, Test function $f(\cdot)$, Number of iterations $K$, a probability distribution $\mathcal{L}$ of an unbiased estimators $\nabla\tilde{U}$ for $\nabla U$

**Output** : $\hat{\pi}_K^{\mathrm{R}}(f) \approx \int f(\theta)\pi(d\theta)$

```
// Run two chains in parallel with consistent Brownian increments
```

**Chain 1:**

**for** $k = 1, \cdots, K$ **do**

  Set random number generator seed to $\mathcal{S}$

  Draw $Z_{2k-1}^{(\gamma/2)} \sim \mathcal{N}(0, \mathrm{Id})$, $Z_{2k}^{(\gamma/2)} \sim \mathcal{N}(0, \mathrm{Id})$

  Set $Z_k^{(\gamma)} = (Z_{2k-1}^{(\gamma/2)} + Z_{2k}^{(\gamma/2)})/\sqrt{2}$

  Draw $\nabla\tilde{U}_k^{(\gamma)}$ from $\mathcal{L}$

  $\theta_k^{(\gamma)} = \theta_{k-1}^{(\gamma)} - \gamma_k \nabla\tilde{U}_k^{(\gamma)}(\theta_{k-1}^{(\gamma)}) + \sqrt{2\gamma_k}Z_k^{(\gamma)}$

$\hat{\pi}_K^{(\gamma)}(f) = \frac{1}{\Gamma_K}\sum_{k=1}^{K}\gamma_{k+1}f(\theta_k^{(\gamma)})$

**Chain 2:**

**for** $k = 1, \cdots, 2K$ **do**

  Set random number generator seed to $\mathcal{S}$

  Draw $Z_k^{(\gamma/2)} \sim \mathcal{N}(0, \mathrm{Id})$

  Draw $\nabla\tilde{U}_k^{(\gamma/2)}$ from $\mathcal{L}$

  $\theta_k^{(\gamma/2)} = \theta_{k-1}^{(\gamma/2)} - \frac{\gamma_{\lceil k/2\rceil}}{2}\nabla\tilde{U}_k^{(\gamma/2)}(\theta_{k-1}^{(\gamma/2)})$
              $+\sqrt{\gamma_{\lceil k/2\rceil}}Z_k^{(\gamma/2)}$

$\hat{\pi}_K^{(\gamma/2)}(f) = \frac{1}{2\Gamma_K}\sum_{k=1}^{2K}\gamma_{\lceil k/2\rceil}f(\theta_k^{(\gamma/2)})$

```
// RR extrapolation.
```

$\hat{\pi}_K^{\mathrm{R}}(f) = 2\hat{\pi}_K^{(\gamma/2)}(f) - \hat{\pi}_K^{(\gamma)}(f)$

---

## 2   Assumptions for the Theoretical Analysis

**Notations:** For $\mathrm{M} \in \mathbb{R}^{d\times d}$, denote by $\mathrm{M} \succ 0$ if and only if $\mathrm{M}$ is a positive definite matrix, and $\mathrm{M} \succeq 0$ if and only if $\mathrm{M}$ is a nonnegative definite matrix. Let $E$ and $F$ be two vector spaces, denote by $E \otimes F$ the tensor product of $E$ and $F$. For all $x \in E$ and $y \in F$ denote by $x \otimes y \in E \otimes F$ the tensor product of $x$ and $y$. Let $n \in \mathbb{N}^*$, denote by $C^n(\mathbb{R}^d)$ the set of $n$ times continuously differentiable functions from $\mathbb{R}^d$ to $\mathbb{R}$. Let $f \in C^n(\mathbb{R}^d)$, denote by $D^n f$ the $n^{\mathrm{th}}$ differential of $f$. Let $f \in C^1(\mathbb{R}^d)$, denote by $\nabla f$ the gradient of $f$. Let $f \in C^2(\mathbb{R}^d)$, denote by $\Delta f$ the Laplacian of $f$. Denote by $\lfloor\cdot\rfloor$ and $\lceil\cdot\rceil$ the floor and ceiling function respectively. For $a, b \in \mathbb{R}$, denote by $a \vee b$ and $a \wedge b$ the maximum and the minimum of $a$ and $b$ respectively.

In this section, we give the full statement of Theorem 1 and Theorem 2, the appropriate conditions which imply these results, and their proof. We begin with two conditions which are common to both Theorems. The first following assumption ensures the stability of the Markov chain produced by SGLD (2) and the diffusion process (1).

**A1.** *There exists a* Lyapunov *function* $V \in C^2(\mathbb{R}^d)$, $V \geq 1$ *with bounded second derivative such that* $\lim_{\|\theta\|\to+\infty} V(\theta) = +\infty$ *and satisfying:*

*(i) Let* $\nabla\tilde{U}$ *be drawn from* $\mathcal{L}$, *almost surely* $\theta \mapsto \nabla\tilde{U}(\theta) \in C^5(\mathbb{R}^d, \mathbb{R}^d)$ *and* $\left\|\nabla\tilde{U}\right\|^2 \leq V$.

*(ii) There exist* $a \geq 0$ *and* $b > 0$ *such that for all* $\theta \in \mathbb{R}^d$, $\|\nabla V(\theta)\|^2 + \|\nabla U(\theta)\|^2 \leq bV(\theta)$ *and* $\langle\nabla V(\theta), \nabla U(\theta)\rangle \geq aV(\theta) - b$.

In a Bayesian inference context where $U$ is the opposite log density of a posterior distribution and is of the form $U(\theta) = -(\sum_{n=1}^{N}\log\mathsf{p}(\mathbf{x}_i|\theta) + \log\mathsf{p}(\theta))$, note that **A**1 holds with the Lyapunov function $V(\theta) = \|\theta\|^\beta + 1$ for $\beta \in (1, 2]$, if there exist $C_1 \geq 0$, $C_2 > 0$ such that for all $\theta \in \mathbb{R}^d$, $\langle\nabla U(\theta), \theta\rangle \geq C_1 \|x\|^\beta - C_2$,

$$\sup_{i\in\{1,\cdots,N\}} \{\|\nabla_\theta U_1(\mathbf{x}_i, \theta)\|^2\} + \|\nabla U_2(\theta)\|^2 \leq C_1(\|\theta\|^\beta + 1),$$

and $\nabla\tilde{U}$ is defined by (3).

Under **A**1-(ii), [1, Theorem 2.2] implies that the process $(\vartheta_t)_{t\geq 0}$ is $\exp(cV)$-uniformly ergodic, for a small constant $c > 0$. Therefore, we have that for all function $f : \mathbb{R}^d \to \mathbb{R}$, $\sup_{x\in\mathbb{R}^d}|f/V^s| < \infty$, for $s \geq 0$, $\pi(f) < +\infty$.

We now make some assumptions on the regularity of the solution of the Poisson equation associated with (1). Define the generator associated with (1) by for all function $h \in C^2(\mathbb{R}^d)$ and all $\theta \in \mathbb{R}^d$,

$$\mathcal{A}h(\theta) = - \langle \nabla U(\theta), \nabla h(\theta) \rangle + \Delta h(\theta) . \tag{S1}$$

**A2** ($q$). *Let $q \in \mathbb{N}$ and $s \geq 0$. For all $f \in C^q(\mathbb{R}^d)$ such that for all $\theta \in \mathbb{R}^d$ and $i \in \{0, \cdots, q\}$, $\left\| D^i f(\theta) \right\| \leq C_f V^s(\theta)$, for $C_f \geq 0$, there exists a unique solution $g \in C^q(\mathbb{R}^d)$ to the Poisson equation $\mathcal{A}g = f - \pi(f)$ satisfying for all $\theta \in \mathbb{R}^d$, $i \in \{0, \cdots, q\}$, $\left\| D^i g(\theta) \right\| \leq C_g V^r(\theta)$ for $C_g, r \geq 0$.*

Let $q \in \mathbb{N}$. [2, Theorem 2] shows that if $U \in C^{q+2}(\mathbb{R}^d)$ and there exist $\beta \in (1, 2]$, $C \geq 0$ such that $\langle \nabla U(x), x \rangle \geq C(\|x\|^\beta - 1)$, then **A2**($q$) holds with $V = 1 + \|\cdot\|$.

## 3 Asymptotic Analysis of SGRRLD

### 3.1 Proof of Theorem 1

We give in this section, the full statement and the proof of Theorem 1. Before that, we make some preliminary observations. Let $(\gamma_k)_{k \geq 1}$ be a sequence of step sizes. Recall that for all $K \geq 1$ and $n \in \mathbb{N}$

$$\Gamma_K^{(n)} = \sum_{k=1}^K \gamma_{k+1}^n , \qquad \Gamma_K = \Gamma_K^{(1)} .$$

We are interested here in the convergence of $\hat{\pi}_K^{\mathrm{R}}(f)$ for a function $f : \mathbb{R}^d \to \mathbb{R}$ and with a sequence of step sizes which satisfies the following assumption.

**A 3.** *The sequence of step sizes $(\gamma_k)_{k \geq 1}$ is nonincreasing and satisfies $\lim_{k \to +\infty} \gamma_k = 0$, $\lim_{k \to +\infty} \Gamma_k = +\infty$.*

Then under **A**1-**A**3, a straightforward application of [3, Theorem 7] implies that almost surely $\lim_{K \to +\infty} \hat{\pi}_K^{\mathrm{R}}(f) = \pi(f)$ as soon as $f : \mathbb{R}^d \to \mathbb{R}$ is a continuous function and $|f/V^s|$ is upper bounded for some exponent $s \geq 0$. under **A**1, [3, Lemma 5] shows that for all $s > 0$, if $\mathbb{E}[V^s(\theta_0)] < +\infty$, then $\sup_{k \geq 0} \mathbb{E}[V^s(\theta_k)] < +\infty$. This upper bound will be used many times in the proofs.

To show Theorem 1, the duplicated diffusion associated with (1) needs to be considered. It is the SDE on $\mathbb{R}^{2d}$ defined by:

$$\begin{cases} \mathrm{d}X_t &= -\nabla U(X_t)\mathrm{d}t + \sqrt{2}\mathrm{d}B_t^{(1)} \\ \mathrm{d}Y_t &= -\nabla U(Y_t)\mathrm{d}t + \sqrt{2}\mathrm{d}B_t^{(2)} , \end{cases} \tag{S2}$$

where $(B_t^{(1)}, B_t^{(2)})_{t \geq 0}$ is a $2d$ dimensional Brownian motion, $(B_t^{(1)})_{t \geq 0}$ and $(B_t^{(2)})_{t \geq 0}$ are $d$-dimensional standard Brownian motion and there exists a $d$-dimensional standard Brownian motion $(\tilde{B}_t)_{t \geq 0}$ independent of $(B_t^{(1)})_{t \geq 0}$ such that

$$B_t^{(2)} = \Sigma^\top B_t^{(1)} + \left(\mathrm{Id} - \Sigma^\top \Sigma\right)^{1/2} \tilde{B}_t , \tag{S3}$$

where $\Sigma \in \mathbb{R}^{d \times d}$ is such that $\mathrm{Id} - \Sigma^\top \Sigma$ is a positive definite matrix. Under **A**1, $\nabla U$ is Lipschitz, therefore (S2) has a unique strong solution $(X_t, Y_t)_{t \geq 0}$. Note that if a probability measure of (S2) is invariant for $(X_t, Y_t)_{t \geq 0}$ then each of its marginal distributions has to be equal to $\pi$. A significant point in the analysis of $\hat{\pi}_K^{\mathrm{R}}$ is the following assumption.

**A4.** *$(X_t, Y_t)_{t \geq 0}$ has a unique invariant measure $\Pi$.*

When $\Sigma$ is invertible, the generator associated with the SDE (S2) is uniformly elliptic and **A**4 is a direct consequence of **A**1, see [4]. In the general case, the generator of (S2) can be hypoelliptic and **A**4 can be more intricate. However, weak assumptions on $U$ which guarantee that **A**4 holds can be found in [5], a particular case being when $U$ is strictly convex [5, Corollary 3.2 (a)].

**A 4** is necessary in the proof because we use that the two sequences $(\theta_{2k}^{(\gamma/2)}, \theta_k^{(\gamma)})_{k \geq 0}$ and $(\theta_{2k-1}^{(\gamma/2)}, \theta_k^{(\gamma)})_{k \geq 0}$ can be viewed as two chains produced by SGLD[1], started at $(\theta_0, \theta_0)$ and $(\theta_1^{(\gamma/2)}, \theta_0)$

respectively, applied to the SDE (S2). Therefore if **A1**-**A3**-**A4** hold, adaptations of the proof of [6, Theorem 1] implies that for every continuous function $h : \mathbb{R}^d \times \mathbb{R}^d \to \mathbb{R}$, $|h| \leq CV^s$ for some $C, s \geq 0$, almost surely we have

$$\lim_{K \to +\infty} \Gamma_K^{-1} \sum_{k=0}^{K} \gamma_{k+1} h(\theta_{2k}^{(\gamma/2)}, \theta_k^{(\gamma)}) = \int_{\mathbb{R}^d \times \mathbb{R}^d} h(x, y) \Pi(\mathrm{d}x, \mathrm{d}y) , \qquad \text{(S4)}$$

and the same statement holds for the chain $(\theta_{2k-1}^{(\gamma/2)}, \theta_k^{(\gamma)})_{k \geq 1}$.

Before giving the full statement of Theorem 1, we need to introduce some notations and definitions. Define for all functions $h_1 \in C^4(\mathbb{R}^d)$ and $h_2 \in C^6(\mathbb{R}^d)$, $\mathcal{G}^{(2)} h_1$ and $\mathcal{G}^{(3)} h_2$ for all $\theta \in \mathbb{R}^d$ by

$$\mathcal{G}^{(2)} h_1(\theta) = \frac{1}{2}(\Delta^2(h_1)(\theta) + \mathbb{E}[D^2 h_1(\theta)\{[\nabla \tilde{U}(\theta)]^{\otimes 2}\}]) - \sum_{i=1}^{d} D^3 h_1(\theta) \left\{\nabla U(\theta) \otimes \mathsf{e}_i^{\otimes 2}\right\} , \quad \text{(S5)}$$

$$\mathcal{G}^{(3)} h_2(\theta) = (6!)^{-1} \mathbb{E}\left[D^6(h_2)(\theta) \left\{Z^{\otimes 6}\right\}\right] - (3!)^{-1} \mathbb{E}\left[D^5 h_2(\theta) \left\{\nabla U(\theta) \otimes Z^{\otimes 4}\right\}\right]$$

$$- (1/3)\mathbb{E}[D^3 h_1(\theta)\{[\nabla \tilde{U}(\theta)]^{\otimes 3}\}] + 3 \sum_{i=1}^{d} \mathbb{E}[D^4 h_1(\theta)\{[\nabla \tilde{U}(\theta)]^{\otimes 2} \otimes \mathsf{e}_i^{\otimes 2}\}] , \quad \text{(S6)}$$

where $\{\mathsf{e}_i\}_{i=1}^d$ stands for the canonical basis of $\mathbb{R}^d$. Assume that **A2**(9) holds and let $f \in C^9(\mathbb{R}^d)$ satisfying for all $\theta \in \mathbb{R}^d$ and $i \in \{0, \cdots, q\}$, $\|D^i f(\theta)\| \leq C_f V^s(\theta)$, for $C_f \geq 0$. Let $g \in C^9(\mathbb{R}^d)$ be the solution of the Poisson equation $\mathcal{A}g = f - \pi(f)$, associated with $f$. Under **A** 1, $\mathcal{G}^{(2)}g \in C^5(\mathbb{R}^d)$ and there exists $C, r > 0$ such that for all $\theta \in \mathbb{R}^d$ and $i \in \{0, \cdots, 5\}$, $\|D^i \{\mathcal{G}^{(2)}g\}(\theta)\| \leq CV^r(\theta)$. Therefore using **A2** again, there exists a unique solution to the Poisson equation $\mathcal{A}G = \mathcal{G}^{(2)}g - \pi(\mathcal{G}^{(2)}g)$, associated with $\mathcal{G}^{(2)}g$, denoted by $G$, such that there exist $\tilde{C}, \tilde{r} > 0$, for all $\theta \in \mathbb{R}^d$ and $i \in \{0, \cdots, 5\}$, $\|D^i G(\theta)\| \leq \tilde{C} V^{\tilde{r}}(\theta)$.

**Theorem S1.** *Let $s \geq 0$ and $f \in C^9(\mathbb{R}^d)$ be a function satisfying for all $\theta \in \mathbb{R}^d$ and $i \in \{0, \cdots, q\}$, $\|D^i f(\theta)\| \leq C_f V^s(\theta)$, for $C_f \geq 0$. Assume **A1**-**A2**(9)-**A3**-**A4**. Let $(\theta_k^{(\gamma)}, \theta_k^{(\gamma/2)})_{k \geq 0}$ be defined by (4)- (5), started at $\theta_0 \in \mathbb{R}^d$ and assume that the relation (8) holds for $\Sigma \in \mathbb{R}^{d \times d}$.*

*a) If $\lim_{K \to +\infty} \Gamma_K^{(3)}/\Gamma_K^{1/2} = 0$, then $\Gamma_K^{1/2}(\hat{\pi}_K^R(f) - \pi(f))$ converges in law as $K$ goes to infinity to a zero-mean Gaussian random variable with variance $\sigma_R^2$ defined by*

$$\sigma_R^2 = 10 \int_{\mathbb{R}^d} \|\nabla g(x)\|^2 \pi(\mathrm{d}x) - 8 \int_{\mathbb{R}^d \times \mathbb{R}^d} \langle \nabla g(x), \Sigma \nabla g(y) \rangle \Pi(\mathrm{d}x, \mathrm{d}y) . \qquad \text{(S7)}$$

*b) If $\lim_{K \to +\infty} \Gamma_K^{(3)}/\Gamma_K^{1/2} = \kappa \in (0, +\infty)$, then $\Gamma_K^{1/2}(\hat{\pi}_K^R(f) - \pi(f))$ converges in law as $K$ goes to infinity to a Gaussian random variable with variance $\sigma_R^2$ and mean $\kappa \mu_R$ where $\sigma_R$ is defined in (S7),*

$$\mu_R = \int_{\mathbb{R}^d} \left\{\mathcal{G}^{(2)}G(\tilde{\theta}) + \mathcal{G}^{(3)}g(\tilde{\theta})\right\} \pi(\mathrm{d}\tilde{\theta}) , \qquad \text{(S8)}$$

*g is the solution of the Poisson equation associated with $f$, $G$ the one associated with $\mathcal{G}^{(2)}g$, $\mathcal{G}^{(2)}G$ and $\mathcal{G}^{(3)}g$ are defined in (S5) and (S6), respectively.*

*c) If $\lim_{K \to +\infty} \Gamma_K^{(3)}/\sqrt{\Gamma_K} = +\infty$, then $(\Gamma_K/\Gamma_k^{(3)})(\hat{\pi}_K^R(f) - \pi(f))$ converges in probability as $K$ goes to infinity to $\mu_R$ given in (S8).*

Before giving the proof of Theorem S1, we make some remarks on the asymptotic variance $\sigma_R$ defined by (S7). Since necessarily the two marginals of $\Pi$ are equal to $\pi$, the Cauchy-Schwarz inequality and the condition $\mathrm{Id} - \Sigma^\top \Sigma$ is definite positive imply that $\sigma_R^2 \geq 2\pi(\|g\|^2)$. On the other hand, when $\Sigma = \mathrm{Id}$, this lower bound is reached, which shows the benefit of choosing consistent Brownian increments. It is also notable that in this case the asymptotic variance is the same as in the case of SGLD.

*Proof of Theorem S1.* Since under **A2**(9), $g \in C^9(\mathbb{R}^d)$, by a 7th order Taylor expansion, denoting by $\Delta\theta_k^\gamma = \theta_{k+1}^{(\gamma)} - \theta_k^{(\gamma)}$ we have for all $k \geq 0$ there exists $s_k^{(\gamma)} \in [0,1]$ such that

$$g(\theta_{k+1}^{(\gamma)}) = g(\theta_k^{(\gamma)}) + \sum_{i=1}^{6} (i!)^{-1} D^i g(\theta_k^{(\gamma)}) \left\{\Delta\theta_k^{(\gamma)}\right\}^{\otimes 7} + (6!)^{-1} D^7 g\left(\theta_k^{(\gamma)} + s_k^{(\gamma)}\theta_k^{(\gamma)}\right) \left\{\Delta\theta_k^{(\gamma)}\right\}^{\otimes 7}$$

$$= g(\theta_{k+1}^{(\gamma)}) + \sum_{i=1}^{6} (i!)^{-1} \sum_{j=0}^{i} \binom{i}{j} 2^{(i-j)/2} \gamma_{k+1}^{(i+j)/2} D^i g(\theta_{k+1}^{(\gamma)}) \left\{ -[\nabla\tilde{U}_{k+1}^{(\gamma)}(\theta_k^{(\gamma)})]^{\otimes j} \otimes [Z_{k+1}^{(\gamma)}]^{\otimes(i-j)} \right\}$$

$$+ (6!)^{-1} D^7 g\left(\theta_k^{(\gamma)} + s_k^{(\gamma)}\theta_k^{(\gamma)}\right) \left\{\Delta\theta_k^{(\gamma)}\right\}^{\otimes 7} .$$

After a change of variables and rearranging the terms, we get

$$g(\theta_{k+1}^{(\gamma)}) = g(\theta_k^{(\gamma)}) + \sum_{i=1}^{12} \gamma_{k+1}^{i/2} \mathcal{D}_k^{(i)} + D^7 g(\theta_k^{(\gamma)} + s_k^{(\gamma)}\Delta\theta_k^{(\gamma)}) \left\{\Delta\theta_k^{(\gamma)}\right\}^{\otimes 7} ,$$

where

$$\mathcal{D}_k^{(i)} = \sum_{j=0}^{\lfloor i/2 \rfloor} \binom{i-j}{j} 2^{(i-2j)/2} D^{i-j} g(\theta_k^{(\gamma)}) \left\{ [-\nabla\tilde{U}_{k+1}^{(\gamma)}(\theta_k^{(\gamma)})]^{\otimes j} \otimes [Z_{k+1}^{(\gamma)}]^{\otimes(i-2j)} \right\} .$$

Let $(\mathcal{F}_i^{(\gamma)})_{i \geq 1}$ be the filtration generated by the sequence $(Z_{2i-1}^{(\gamma/2)}, Z_{2i}^{(\gamma/2)}, Z_i^{(\gamma)}, \nabla\tilde{U}_{2i-1}^{(\gamma/2)}, \nabla\tilde{U}_{2i}^{(\gamma/2)}, \nabla\tilde{U}_i^{(\gamma)})_{i \geq 1}$ and $\theta_0^{(\gamma/2)}$. Introducing the conditional expectation of $\mathcal{D}_k^{(i)}$ for all $i \in \{1, \cdots, 12\}$, we have

$$g(\theta_{k+1}^{(\gamma)}) = g(\theta_k^{(\gamma)}) + 2^{1/2} \gamma_{k+1}^{1/2} Dg(\theta_k^{(\gamma)})[Z_{k+1}^{(\gamma)}] + \gamma_{k+1} \mathcal{A}g(\theta_k^{(\gamma)}) + \gamma_{k+1}^2 \mathcal{G}^{(2)} g(\theta_k^{(\gamma)})$$

$$+ \gamma_{k+1}^3 \mathcal{G}^{(3)} g(\theta_k^{(\gamma)}) + \sum_{i=2}^{12} \gamma_{k+1}^{i/2} \mathcal{E}_k^{(i)} + (6!)^{-1} D^7 g\left(\theta_k^{(\gamma)} + s_k^{(\gamma)}\theta_k^{(\gamma)}\right) \left\{\Delta\theta_k^{(\gamma)}\right\}^{\otimes 7} ,$$

where $\mathcal{A}g, \mathcal{G}^{(2)}g, \mathcal{G}^{(3)}g$ are defined in (S1), (S5), (S6) respectively and $\mathcal{E}_k^{(i)} = \mathcal{D}_k^{(i)} - \mathbb{E}[\mathcal{D}_k^{(i)} | \mathcal{F}_k]$. Therefore since $\mathcal{A}g = f - \pi(f)$, we have for all $K \geq 1$,

$$\sum_{k=1}^{K} \gamma_{k+1} \left\{ f(\theta_k^{(\gamma)}) - \pi(f) \right\} = g(\theta_{k+1}^{(\gamma)}) - g(\theta_1^{(\gamma)}) - \sum_{k=1}^{K} 2^{1/2} \gamma_{k+1}^{1/2} Dg(\theta_k^{(\gamma)}) Z_{k+1}^{(\gamma)} - \sum_{k=1}^{K} \gamma_{k+1}^2 \mathcal{G}^{(2)} g(\theta_k^{(\gamma)})$$

$$- \sum_{k=1}^{K} \gamma_{k+1}^3 \mathcal{G}^{(3)} g(\theta_k^{(\gamma)}) - \sum_{k=1}^{K} \sum_{i=2}^{12} \gamma_{k+1}^{i/2} \mathcal{E}_k^{(i)} - (6!)^{-1} D^7 g\left(\theta_k^{(\gamma)} + s_k^{(\gamma)}\theta_k^{(\gamma)}\right) \left\{\Delta\theta_k^{(\gamma)}\right\}^{\otimes 7} . \quad \text{(S9)}$$

Similarly for $(\theta^{(\gamma/2)})_{k \geq 0}$, but conditioning this time by the filtration $(\mathcal{F}_i^{(\gamma/2)})_{i \geq 0}$ generated by the sequence $(Z_i^{(\gamma/2)}, \nabla\tilde{U}_i^{(\gamma/2)})_{i \geq 1}$ and $\theta_0^{(\gamma/2)}$, we have for all $k \geq 1$,

$$g(\theta_{k+1}^{(\gamma/2)}) = g(\theta_k^{(\gamma/2)}) + \gamma_{k+1}^{1/2} Dg(\theta_k^{(\gamma/2)}) Z_{k+1}^{(\gamma/2)} + (\gamma_{k+1})(\mathcal{A}g(\theta_k^{(\gamma/2)})) + (\gamma_{k+1}^2/4)\mathcal{G}^{(2)} g(\theta_k^{(\gamma/2)})$$

$$+ (\gamma_{k+1}^3/8)\mathcal{G}^{(3)} g(\theta_k^{(\gamma/2)}) + \sum_{i=1}^{12} (\gamma_{k+1}/2)^{i/2} \mathcal{K}_k^{(i)} + (6!)^{-1} D^7 g\left(\theta_k^{(\gamma/2)} + s_k^{(\gamma/2)}\Delta\theta_{k+1}^{(\gamma/2)}\right) \left\{\Delta\theta_{k+1}^{(\gamma/2)}\right\}^{\otimes 7} .$$

where $s_k^{(\gamma/2)} \in [0,1]$, $\mathcal{K}_k^{(i)} = \mathcal{H}_k^{(i)} - \mathbb{E}[\mathcal{H}_k^{(i)} | \mathcal{F}_i^{(\gamma/2)}]$ and

$$\mathcal{H}_k^{(i)} = \sum_{j=0}^{\lfloor i/2 \rfloor} \binom{i-j}{j} 2^{(i-2j)/2} D^{i-j} g(\theta_k^{(\gamma)}) \left\{ -[\nabla\tilde{U}_{k+1}^{(\gamma)}(\theta_k^{(\gamma)})]^{\otimes j} \otimes [Z_{k+1}^{(\gamma)}]^{\otimes(i-2j)} \right\} .$$

This equality implies that for all $K \geq 1$,

$$\sum_{k=1}^{2K} \gamma_{k+1} \left\{ f(\theta_k^{(\gamma/2)}) - \pi(f) \right\} = g(\theta_{2K+1}^{(\gamma/2)}) - g(\theta_1^{(\gamma/2)}) - \sum_{k=1}^{2K} \gamma_{k+1}^{1/2} Dg(\theta_k^{(\gamma/2)}) Z_{k+1}^{(\gamma/2)} \quad \text{(S10)}$$

$$- \sum_{k=1}^{2K} \left\{ (\gamma_{k+1}^2/4)\mathcal{G}^{(2)} g(\theta_k^{(\gamma/2)}) + (\gamma_{k+1}^3/8)\mathcal{G}^{(3)} g(\theta_k^{(\gamma/2)}) + \sum_{i=2}^{12} (\gamma_{k+1}/2)^{i/2} \mathcal{E}_k^{(i)} \right\}$$

$$- (6!)^{-1} D^7 g\left( \theta_k^{(\gamma/2)} + s_k^{(\gamma/2)} \Delta\theta_k^{(\gamma/2)} \right) \left\{ \Delta\theta_k^{(\gamma/2)} \right\}^{\otimes 7} .$$

Combining (S9) and (S10), we get

$$\Gamma_K \left\{ \hat{\pi}_K^{\mathrm{R}}(f) - \pi(f) \right\} = 2(g(\theta_{2K+1}^{(\gamma/2)}) - g(\theta_1^{(\gamma/2)})) - g(\theta_{K+1}^{(\gamma)}) - g(\theta_1^{(\gamma)})$$

$$- \sum_{k=1}^{K} \left\{ \gamma_{k+1}^{1/2} \mathcal{M}_k + \gamma_{k+1}^2 \mathcal{B}_k^{(2)} + \gamma_{k+1}^3 \mathcal{B}_k^{(3)} + \mathcal{N}_k + \mathcal{R}_k \right\} , \quad \text{(S11)}$$

where

$$\mathcal{M}_k = 2 \left\{ Dg(\theta_{2k-1}^{(\gamma/2)})[Z_{2k}^{(\gamma/2)}] + Dg(\theta_{2k}^{(\gamma/2)})[Z_{2k+1}^{(\gamma/2)}] \right\} - 2^{1/2} Dg(\theta_k^{(\gamma)})[Z_{k+1}^{(\gamma)}] \quad \text{(S12)}$$

$$\mathcal{B}_k^{(i)} = 2^{1-i} \left\{ \mathcal{G}^{(i)} g(\theta_{2k-1}^{(\gamma/2)}) + \mathcal{G}^{(i)} g(\theta_{2k}^{(\gamma/2)}) \right\} - \mathcal{G}^{(i)} g(\theta_k^{(\gamma)}) \quad \text{for } i = 2, 3 \quad \text{(S13)}$$

$$\mathcal{N}_k = \sum_{i=2}^{6} \left\{ 2(\gamma_{k+1}/2)^{i/2} (\mathcal{K}_{2k-1}^{(i)} + \mathcal{K}_{2k}^{(i)}) - \gamma_{k+1}^{i/2} \mathcal{E}_k^{(i)} \right\} \quad \text{(S14)}$$

$$\mathcal{R}_k = \sum_{i=7}^{12} \left\{ 2(\gamma_{k+1}/2)^{i/2} (\mathcal{K}_{2k-1}^{(i)} + \mathcal{K}_{2k}^{(i)}) - \gamma_{k+1}^{i/2} \mathcal{E}_k^{(i)} \right\} - D^7 g(\theta_k^{(\gamma)} + s_k^{(\gamma)} \Delta\theta_k^{(\gamma)}) \left\{ \Delta\theta_k^{(\gamma)} \right\}^{\otimes 7}$$

$$+ 2(6!)^{-1} D^7 g(\theta_{2k-1}^{(\gamma/2)} + s_{2k-1}^{(\gamma/2)} \Delta\theta_{2k-1}^{(\gamma/2)}) \left\{ \Delta\theta_{2k-1}^{(\gamma/2)} \right\}^{\otimes 7}$$

$$+ 2(6!)^{-1} D^7 g(\theta_{2k}^{(\gamma/2)} + s_{2k}^{(\gamma/2)} \Delta\theta_{2k}^{(\gamma/2)}) \left\{ \Delta\theta_{2k}^{(\gamma/2)} \right\}^{\otimes 7} . \quad \text{(S15)}$$

First under **A2**(9), $g \leq C_g V^r$, therefore using **A1** and **A3** and [3, Theorem 7] we get

$$\lim_{K \to +\infty} \Gamma_K^{-1/2} \mathbb{E} \left[ \left| g(\theta_{2K+1}^{(\gamma/2)}) - g(\theta_1^{(\gamma/2)}) \right| \right] = 0 ,$$

which implies that $\Gamma_K^{-1/2}(g(\theta_{2K+1}^{(\gamma/2)}) - g(\theta_1^{(\gamma/2)}))$ goes to 0 in probability. Using the same reasoning, we have that $\Gamma_K^{-1/2}(g(\theta_{K+1}^{(\gamma)}) - g(\theta_1^{(\gamma)}))$ goes to 0 as well. The proof then consists in controlling the weighted sums of each term appearing in (S11). The two leading contributions are:

1. $\sum_{k=1}^{K} \gamma_{k+1} \mathcal{M}_k$ is the fluctuation term, which converges in law to a zero-mean Gaussian random variable if it is scaled by $\Gamma_K^{1/2}$. It is the content of Lemma S2.

2. $\sum_{k=1}^{K} \gamma_{k+1}^2 \mathcal{B}_k^{(2)} + \gamma_{k+1}^3 \mathcal{B}_k^{(3)}$ is the bias term. It is shown in Lemma S3 and b) below that, divided by $\Gamma_K^{(3)}$, this term converges in probability to a constant. Note that this is the main difference between the convergence of SGLD and SGRRLD. Indeed, in the case of SGLD the bias term converges at the rate $\Gamma_K^{(2)}$, see [3, Theorem 8].

As regards to the other terms in (S11), it is shown in Lemma S5 that they are negligible. $\qquad \square$

**Lemma S2.** *Under the assumptions of Theorem 1, $\Gamma_K^{-1/2} \sum_{k=1}^{K} \gamma_{k+1}^{1/2} \mathcal{M}_k$ converges in law to a zero-mean Gaussian random variable with variance $\sigma_{\mathrm{R}}^2$ where $(\mathcal{M}_k)_{k \geq 1}$ and $\sigma_{\mathrm{R}}^2$ are defined in (S12) and (S7) respectively.*

*Proof.* Denote by $\xi_{k,K} = (\gamma_{k+1}/\Gamma_K)^{1/2}\mathcal{M}_k$ for $K \geq 1$ and $k \in \{1, \cdots, K\}$. The proof consists in applying a CLT for the arrays of $(\mathcal{F}_k^{(\gamma)})_{k\geq 0}$-martingale increments $(\xi_{1,K}, \cdots, \xi_{k,K})_{K\geq 1}$. By [7, Corollary 3.1], it is sufficient to show that almost surely

$$\lim_{K\to+\infty} \sum_{k=1}^{K} \mathbb{E}\left[\xi_{k,K}^2 \,\middle|\, \mathcal{F}_{k-1}^{(\gamma)}\right] = \sigma_{\mathrm{R}}^2 \tag{S16}$$

$$\lim_{K\to+\infty} \sum_{k=1}^{K} \mathbb{E}\left[|\xi_{k,K}|^3 \,\middle|\, \mathcal{F}_{k-1}^{(\gamma)}\right] = 0 \ . \tag{S17}$$

Let us first show (S16). By definition (S12) of $(\mathcal{M}_k)_{k\geq 1}$ and (8), for all $K \geq 1$, $k \in \{1, \cdots, K\}$, we have

$$\mathbb{E}\left[\xi_{k,K}^2 \,\middle|\, \mathcal{F}_k^{(\gamma)}\right] = \frac{\gamma_{k+1}}{\Gamma_K}\left\{4\left\|Dg(\theta_{2k-1}^{(\gamma/2)})\right\|^2 + 4\left\|Dg(\theta_{2k}^{(\gamma/2)})\right\|^2 + 2\left\|Dg(\theta_k^{(\gamma)})\right\|^2 \right.$$
$$\left. 4\left\langle Dg(\theta_{2k-1}^{(\gamma/2)}), \Sigma Dg(\theta_k^{(\gamma)})\right\rangle + 4\left\langle Dg(\theta_{2k}^{(\gamma/2)}), \Sigma Dg(\theta_k^{(\gamma)})\right\rangle\right\} \ . \tag{S18}$$

Since $(\theta_k^{(\gamma/2)})_{k\geq 0}$ is a Markov chain produced by the SGLD applied to (1) with the sequence of step sizes $(\eta_k)_{k\geq 1}$ defined by $\eta_{2k-1} = \gamma_k/2$ and $\eta_{2k} = \gamma_k/2$, and **A1**-**A2**(9)-**A3** are assumed, [3, Theorem 7] can be applied and almost surely

$$\lim_{K\to+\infty} \sum_{k=1}^{K}(\gamma_{k+1}/\Gamma_K)\left\{\left\|Dg(\theta_{2k-1}^{(\gamma/2)})\right\|^2 + \left\|Dg(\theta_{2k}^{(\gamma/2)})\right\|^2\right\} = 2\int_{\mathbb{R}^d}\|Dg(x)\|^2\,\pi(\mathrm{d}x) \ . \tag{S19}$$

The same result can be applied to the sequence $(\theta_k^{(\gamma)})_{k\geq 0}$ which implies that

$$\lim_{K\to+\infty} \sum_{k=1}^{K}(\gamma_{k+1}/\Gamma_K)\left\|Dg(\theta_k^{(\gamma)})\right\|^2 = \int_{\mathbb{R}^d}\|Dg(x)\|^2\,\pi(\mathrm{d}x) \ . \tag{S20}$$

For the other terms, by **A1**-**A2**(9)-**A3** and (S4), we have for $i = 0, 1$

$$\lim_{K\to+\infty} \sum_{k=1}^{K}(\gamma_{k+1}/\Gamma_K)\left\langle Dg(\theta_{2k-i}^{(\gamma/2)}), \Sigma Dg(\theta_k^{(\gamma)})\right\rangle = \int_{\mathbb{R}^d}\langle Dg(x), \Sigma Dg(y)\rangle\,\Pi(\mathrm{d}x, \mathrm{d}y) \ . \tag{S21}$$

Combining (S19)-(S20)-(S21) in (S18) shows (S16). We now deal with showing (S17). By Hölder's inequality, **A2**, we have

$$\mathbb{E}\left[|\xi_{k,K}|^3 \,\middle|\, \mathcal{F}_k^{(\gamma)}\right] \leq C(\gamma_{k+1}/\Gamma_K)^{3/2}\left\{\left\|Dg(\theta_{2k-1}^{(\gamma/2)})\right\|^3 + \left\|Dg(\theta_{2k}^{(\gamma/2)})\right\|^3 + \left\|Dg(\theta_k^{(\gamma)})\right\|^3\right\}$$
$$\leq C(\gamma_{k+1}/\Gamma_K)^{3/2}\left\{V^{3r}(\theta_{2k-1}^{(\gamma/2)}) + V^{3r}(\theta_{2k}^{(\gamma/2)}) + V^{3r}(\theta_k^{(\gamma)})\right\} \ .$$

By [3, Theorem 7], almost surely,

$$\sup_{K\geq 1}(\Gamma_K^{(3/2)})^{-1} \sum_{k=1}^{K+1} \gamma_{k+1}^{3/2}\left\{V^{3r}(\theta_{2k-1}^{(\gamma/2)}) + V^{3r}(\theta_{2k}^{(\gamma/2)}) + V^{3r}(\theta_k^{(\gamma)})\right\} < +\infty \ .$$

Therefore, using that $(\gamma_k)_{k\geq 1}$ is nonincreasing, we have almost surely, for all $K \geq 2$,

$$\sum_{k=1}^{K} \mathbb{E}\left[|\xi_{k,K}|^3 \,\middle|\, \mathcal{F}_k^{(\gamma)}\right] \leq C\Gamma_K^{(3/2)}/\Gamma_K^{3/2} \leq C\Gamma_K^{-1/2} \ ,$$

which concludes the proof of (S17) since $\lim_{K\to+\infty} \Gamma_K = +\infty$. □

**Lemma S3.** *Under the assumptions of Theorem 1, the following statements hold:*

a) *If* $\lim_{K\to+\infty} \Gamma_K^{(3)} = +\infty$. *Then in probability,*

$$\lim_{K\to+\infty} (\Gamma_{K+1}^{(3)})^{-1} \sum_{k=1}^{K} \gamma_{k+1}^2 \mathcal{B}_k^{(2)} = -(1/2)\mu_{\mathrm{R}}^{(2)} \, ,$$

*where* $\mu_{\mathrm{R}}^{(2)} = \pi(\mathcal{G}^{(2)}G)$, *and* $G$ *is the solution of the Poisson equation associated with* $\mathcal{G}^{(2)}g$, $\mathcal{A}G = \mathcal{G}^{(2)}g - \pi(\mathcal{G}^{(2)}g)$.

b) *If* $\lim_{K\to+\infty} \Gamma_K^{(3)} < +\infty$. *Then in probability,*

$$\lim_{K\to+\infty} (\Gamma_K)^{-1/2} \sum_{k=1}^{K} \gamma_{k+1}^2 \mathcal{B}_k^{(2)} = 0 \, .$$

*Proof.* a) Under **A**1 and **A**2, $\mathcal{G}^{(2)}g$ is integrable w.r.t. $\pi$. Introducing $\pi(\mathcal{G}^{(2)}g)$, we have

$$(\Gamma_K^{(3)})^{-1} \sum_{k=1}^{K} \gamma_{k+1}^2 \mathcal{B}_k^{(2)} = (\Gamma_K^{(3)})^{-1}(A_K^{(1)} + A_K^{(2)}) \, , \qquad (\mathrm{S}22)$$

where

$$A_K^{(1)} = 2^{-1} \sum_{k=1}^{2K} \eta_{k+1}^2 \left\{ \mathcal{G}^{(2)}g(\theta_k^{(\gamma/2)}) - \pi(\mathcal{G}^{(2)}g) \right\}$$

$$A_K^{(2)} = \sum_{k=1}^{K} \gamma_{k+1}^2 \left\{ \mathcal{G}^{(2)}g(\theta_k^{(\gamma)}) - \pi(\mathcal{G}^{(2)}g) \right\} \, ,$$

the sequence $(\eta_k)_{k\geq 1}$ is defined by $\eta_{2k-1} = \gamma_k/2$ and $\eta_{2k} = \gamma_k/2$. As mentioned before the statement of Theorem S1, using again Under **A**1 and **A**2, we verify that $\mathcal{G}^{(2)}g$ satisfies **A**2(5). Therefore the solution $\phi^{(2)}$ of the Poisson equation $\mathcal{A}G = \mathcal{G}^{(2)} - \pi(\mathcal{G}^{(2)})$ belongs to $C^5(\mathbb{R}^d)$ and there exists $\tilde{r} \geq 0$ such that for all $i \in \{1, \cdots, 5\}$, $x \in \mathbb{R}^d$, $\left\| D^i G(x) \right\| \leq CV^{\tilde{r}}(x)$. Therefore by an adaptation of the proof of [8, Theorem V.3] for SGLD, we have that in probability $\lim_{K\to+\infty} (\Gamma_K^{(2)}/\Gamma_K^{(3)})A_K^{(1)} = \mu_{\mathrm{R}}^{(2)}/2$ and $\lim_{K\to+\infty} (\Gamma_K^{(2)}/\Gamma_K^{(3)})A_K^{(2)} = -\mu_{\mathrm{R}}^{(2)}$, which concludes the proof of the first point.

b) The proof of the second point follows the same line and is omitted.

$\qquad\qquad\qquad\qquad\qquad\qquad\qquad\qquad\qquad\qquad\qquad\qquad\qquad\qquad\qquad\qquad \square$

**Lemma S4.** *Under the assumptions of Theorem 1, the following statements hold:*

a) *If* $\lim_{K\to+\infty} \Gamma_K^{(3)} = +\infty$. *Then almost surely,*

$$\lim_{K\to+\infty} (\Gamma_K^{(3)})^{-1} \sum_{k=1}^{K} \gamma_{k+1}^3 \mathcal{B}_k^{(3)} = \pi(\mathcal{G}^{(3)}g) \, .$$

.

b) *If* $\lim_{K\to+\infty} \Gamma_K^{(3)} < +\infty$. *Then almost surely,*

$$\lim_{K\to+\infty} (\Gamma_K)^{-1/2} \sum_{k=1}^{K} \gamma_{k+1}^3 \mathcal{B}_k^{(3)} = 0 \, .$$

*Proof.* The proof is a straightforward application of [3, Theorem 7] to the sequence of weights $(\gamma_k^3)_{k\geq 1}$, see [3, Remark 3]. $\qquad\qquad\qquad\qquad\qquad\qquad\qquad\qquad\qquad \square$

**Lemma S5.** *Under the assumption of Theorem 1, the following limit holds in probability*

$$\lim_{K\to+\infty} \Xi_K^{-1} \sum_{k=1}^{K} \{ \mathcal{N}_k + \mathcal{R}_k \} = 0 \, ,$$

*where* $\Xi_K = \Gamma_K \vee \Gamma_K^{1/2}$.

*Proof.* We consider each term appearing in the definition (S14) and (S15) of $(\mathcal{N}_k)_{k\geq 1}$ and $(\mathcal{R}_k)_{k\geq 0}$. Let us deal with the first term of $(\mathcal{N}_k)_{k\geq 1}$, the proof for the other terms follows the same line and is omitted. By **A1-A2**, there exist $C, p \geq 0$ such that

$$\Gamma_K^{-1} \sum_{k=1}^{K+1} \gamma_{k+1}^2 \mathbb{E}\left[\left|\mathcal{E}_k^{(2)}\right|^2\right] \leq C\Gamma_K^{-1} \sum_{k=1}^{K+1} \gamma_{k+1}^2 \mathbb{E}\left[V^p(\theta_k^{(\gamma/2)})\right]$$

$$\leq C\Gamma_K^{-1}\left(1 + \sum_{k=1}^{K} \Gamma_{k+1}^{-1}\left\{\gamma_{k+1} - \gamma_{k+2}\right\}\right).$$

By Kronecker's lemma, we get

$$\lim_{K\to+\infty} \Gamma_K^{-1} \sum_{k=1}^{K+1} \gamma_{k+1}^2 \mathbb{E}\left[\left|\mathcal{E}_k^{(2)}\right|^2\right] = 0.$$

Since $(\mathcal{E}_k^{(2)})_{k\geq 1}$ is a sequence of $(\mathcal{F}_{k+1}^{(\gamma/2)})_{k\geq 0}$-martingale increment, it holds that in probability,

$$\lim_{K\to+\infty} \Gamma_K^{-1/2} \sum_{k=1}^{K+1} \gamma_{k+1} \mathcal{E}_k^{(2)} = 0.$$

It can be proved in a similar manner that the term involving $(\mathcal{K}_k^{(2)})_{k\geq 1}$ converges to 0 as well. □

## 3.2 Discussion on Theorem 1

If $(\gamma_k)_{k\geq 1}$ is of the form $\gamma_k = \gamma_1 k^{-\alpha}$ for $\alpha \in (0,1]$ then for $K \geq 1$ large enough, $\Gamma_K^{1/2} = \mathcal{O}(K^{(1-\alpha)/2})$ and $\Gamma_K^{(3)} = \mathcal{O}(K^{1-3\alpha})$. By Theorem S1, $\hat{\pi}_K^{\mathrm{R}}(f)$ converges to $\pi(f)$ at a rate of convergence of order $\Gamma_K^{1/2} \wedge (\Gamma_K/\Gamma_K^{(3)})$, which corresponds in this case to $\mathcal{O}(K^{-((1-\alpha)/2)\wedge(2\alpha)})$.

In the case $\alpha = 1/5$, then $\Gamma_K^{1/2} \sim (5\gamma_1/4)^{1/2} K^{2/5}$, $\Gamma_K^{(3)} \sim (5\gamma_1^3/2)K^{2/5}$, as $K$ goes to infinity. Therefore, $\lim_{K\to+\infty} \Gamma_K^{(3)}/\Gamma_K^{1/2} = 5^{1/2}\gamma_1^{5/2}$. By Theorem S1-b), we get that $n^{2/5}(\hat{\pi}_K^{\mathrm{R}}(f) - \pi)$ converges in law to a Gaussian distribution with mean $2\gamma_1^2 \mu_{\mathrm{R}}$ and variance $(4/(5\gamma_1))\sigma_{\mathrm{R}}^2$. The second moment of this distribution is $4\gamma_1^4 \mu_{\mathrm{R}}^2 + (4/(5\gamma_1))\sigma_{\mathrm{R}}^2$. An easy computation shows that this quantity is minimal when $\gamma_1 = (\sigma_{\mathrm{R}}^2/(20\mu_{\mathrm{R}}^2))^{1/5}$.

# 4 Non-asymptotic Analysis of SGRRLD

In this section, we give the full statement of Theorem 2 and the conditions which imply this result. Under the assumption that $\gamma_k$ is small enough for large $k$ and **A1** we could adapt the proof [3, Lemma 5] to show that for all $r > 0$, $\sup_{k\geq 0} \mathbb{E}[V^r(\theta_k)] < +\infty$, but to clarify the presentation we make the following assumption.

**A5.** *The sequence $(\gamma_k)_{k\geq 1}$ is nonincreasing, $\lim_{K\to+\infty} \Gamma_K = +\infty$, for some $K_1 \geq 1$, $\gamma_{K_1} \leq 1$ and for all $r > 0$, $\sup_{k\geq 0} \mathbb{E}[V^r(\theta_k^{(\gamma)})] < +\infty$, $\sup_{k\geq 0} \mathbb{E}[V^r(\theta_k^{(\gamma/2)})] < +\infty$.*

**Theorem S6.** *Let $s \geq 0$ and $f \in C^9(\mathbb{R}^d)$ be a function satisfying for all $\theta \in \mathbb{R}^d$ and $i \in \{0, \cdots, q\}$, $\left\|D^i f(\theta)\right\| \leq C_f V^s(\theta)$, for $C_f \geq 0$. Let $(\theta_k^{(\gamma)}, \theta_k^{(\gamma/2)})_{k\geq 0}$ be defined by (4)- (5), started at $\theta_0 \in \mathbb{R}^d$. Assume **A1-A2** and **A5**. Then there exists $C \geq 0$ such that for all $K \in \mathbb{N}$, $K \geq 1$:*

$$\text{BIAS:} \qquad \left|\mathbb{E}\left[\hat{\pi}_K^{\mathrm{R}}(f) - \pi(f)\right]\right| \leq (C/\Gamma_K)\left\{\mathsf{m}_3\,\Gamma_K^{(3)} + 1\right\}$$

$$\text{MSE:} \qquad \mathbb{E}\left[\left\{\hat{\pi}_K^{\mathrm{R}}(f) - \pi(f)\right\}^2\right] \leq C\{(\mathsf{m}_3\Gamma_K^{(3)}/\Gamma_K)^2 + 1/\Gamma_K\},$$

*where $\mathsf{m}_3 = \mathbb{E}[\|\nabla\tilde{U}_1\|^3]$.*

*Proof.* **Proof for the Bias:** We use the decomposition given in (S11), which implies that taking the expectation

$$\Gamma_K \left| \mathbb{E}\left[\hat{\pi}_K^{\mathrm{R}}(f) - \pi(f)\right]\right| = \mathbb{E}\left[2(g(\theta_{2K+1}^{(\gamma/2)}) - g(\theta_1^{(\gamma/2)})) - g(\theta_{K+1}^{(\gamma)}) - g(\theta_1^{(\gamma)})\right]$$
$$- \sum_{k=1}^{K} \mathbb{E}\left[\gamma_{k+1}^2 \mathcal{B}_k^{(2)} + \gamma_{k+1}^3 \mathcal{B}_k^{(3)} + \mathcal{R}_k\right] , \quad \text{(S23)}$$

Using **A2**(9), **A5** we get there exists $C \geq 0$,

$$\sup_{K \geq 1} \mathbb{E}\left[\left| g(\theta_{2K+1}^{(\gamma/2)}) - g(\theta_1^{(\gamma/2)}) + g(\theta_{K+1}^{(\gamma)}) - g(\theta_1^{(\gamma)})\right|\right] < C \quad \text{(S24)}$$

$$\sum_{k=1}^{K} \gamma_{k+1}^3 \mathbb{E}\left[\left|\mathcal{B}_k^{(3)}\right|\right] \leq C\mathsf{m}_3 \Gamma_K^{(3)} \qquad \sum_{k=1}^{K} \mathbb{E}\left[|\mathcal{R}_k|\right] \leq C\Gamma_K^{(7/2)} . \quad \text{(S25)}$$

It remains to bound the terms involving $\mathcal{B}_k^{(2)}$. Introducing the integral of $\mathcal{G}^{(2)}g$ w.r.t. $\pi$, as it is done in Lemma S4, the solution of the Poisson equation $G$ associated with $\mathcal{G}^{(2)}g$ and using a 5-th order Taylor expansion of $G(\theta_{k+1}^{(\gamma)})$ at $(\theta_k^{(\gamma)})_{k \geq 0}$, as done for (S11), we get

$$\sum_{k=1}^{K+1} \gamma_{k+1}^2 \mathbb{E}\left[\mathcal{G}^{(2)}g(\theta_k^{(\gamma)}) - \pi(\mathcal{G}^{(2)}g)\right] = \sum_{k=1}^{K+1} \gamma_{k+1} \mathbb{E}\left[G(\theta_{k+1}^{(\gamma)}) - G(\theta_k^{(\gamma)}) + \gamma_{k+1}^2 A(\theta_k^{(\gamma)}) + R_k\right] ,$$

where almost surely $A(\theta) \leq CV^p(\theta)$ and $R_k \leq C\gamma_{k+1}^{5/2} V^p(\theta_k^{(\gamma)})$, for $C, p \geq 0$. Using again **A2**(9), **A5** and a summation by parts, we have

$$\left| \sum_{k=1}^{K+1} \gamma_{k+1}^2 \mathbb{E}\left[\mathcal{G}^{(2)}g(\theta_k^{(\gamma)}) - \pi(\mathcal{G}^{(2)}g)\right]\right| \leq C(1 + \Gamma_K^{(3)}) . \quad \text{(S26)}$$

Similarly, we show that

$$\left| \sum_{k=1}^{K+1} \gamma_{k+1}^2 \mathbb{E}\left[\mathcal{G}^{(2)}g(\theta_k^{(\gamma)}) - \pi(\mathcal{G}^{(2)}g)\right]\right| \leq C(1 + \Gamma_K^{(3)}) . \quad \text{(S27)}$$

Combining (S24)-(S25)-(S26)-(S27) in (S23) concludes the proof.

**Proof for the MSE:** Using (S11), we have there exists $C \geq 0$ such that

$$\Gamma_K^2 \mathbb{E}\left[\left(\hat{\pi}_K^{\mathrm{R}}(f) - \pi(f)\right)^2\right] \leq C \left( \mathbb{E}\left[\left(2(g(\theta_{2K+1}^{(\gamma/2)}) - g(\theta_1^{(\gamma/2)})) - g(\theta_{K+1}^{(\gamma)}) - g(\theta_1^{(\gamma)})\right)^2\right]\right.$$

$$+ \mathbb{E}\left[\left(\sum_{k=1}^{K} \left\{\gamma_{k+1}^{1/2} \mathcal{M}_k\right\}^2\right)\right] + \mathbb{E}\left[\left(\sum_{k=1}^{K} \gamma_{k+1}^2 \mathcal{B}_k^{(2)}\right)^2\right] + \mathbb{E}\left[\left(\sum_{k=1}^{K} \gamma_{k+1}^3 \mathcal{B}_k^{(3)}\right)^2\right]$$

$$\left. + \mathbb{E}\left[\left(\sum_{k=1}^{K} \mathcal{N}_k\right)^2\right] + \mathbb{E}\left[\left(\sum_{k=1}^{K} \mathcal{R}_k\right)^2\right]\right) .$$

As for the bias, we need to control each term. The main difference is the terms involving the martingales increments $\mathcal{M}_k$ and $\mathcal{N}_k$, which can be easily bounded using **A2**(9)-**A5** by

$$\mathbb{E}\left[\left(\sum_{k=1}^{K} \left\{\gamma_{k+1}^{1/2} \mathcal{M}_k\right\}^2\right)\right] + \mathbb{E}\left[\left(\sum_{k=1}^{K} \mathcal{N}_k\right)^2\right] \leq C\Gamma_K$$

For the others, the proof follows from a straightforward modification of the proof for the bias. $\quad \square$

Figure S1: Bias and MSE of SGRRLD for decreasing step size $\gamma_k \propto k^{-\alpha}$, with $\alpha \in \{0.1, 0.2, 0.33, 0.5\}$.

## 5 Stochastic Gradient Hamiltonian Monte Carlo

We present here two algorithms based on the second order Langevin dynamics associated with $\pi$, defined as the SDE on $\mathbb{R}^{2d}$:

$$\begin{cases} \mathrm{d}\vartheta_t &= \rho_t \mathrm{d}t \\ \mathrm{d}\rho_t &= -\omega\rho_t - \nabla U(\vartheta_t)\mathrm{d}t + \sqrt{2\rho}\mathrm{d}B_t \ , \end{cases} \tag{S28}$$

where $\omega \in \mathbb{R}_+^*$ is the friction parameter. It can be shown, see [9], that this dynamics has a stationary distribution with density w.r.t. the Lebesgue measure proportional to $(\theta, r) \mapsto \mathrm{e}^{-U(\theta) - \|r\|^2/2}$.

It has been proposed in [9] to use an Euler discretization for (S28), where similarly to SGLD, the gradient is replaced by a noisy estimate. The full algorithm is given in Algorithm 3 and its RR extrapolation in Algorithm 4.

It has been shown in [10] that the SGHMC with the Euler integrator is a first order integrator. They also proposed a symmetric splitting integrator for (S28) and proved that it is a second order integrator. The full algorithm is presented in Algorithm 5.

## 6 Additional experiments

We present an experiment which shows the optimal rates of convergence that we have derived in Theorem 2 for a decreasing sequence of step sizes of the form $\gamma_k \propto k^{-\alpha}$ for all $k \geq 1$, with $\alpha \in (0, 1)$. From the bounds given in Theorem 2, the optimal sequence for the bias is of the form $\gamma_k = \gamma_b^\star k^{-1/3}$ and $\gamma_k = \gamma_M^\star k^{-1/5}$ for the MSE. We run a first experiment to determine the constants $\gamma_b^\star$ and $\gamma_M^\star$ with 20000 iterations. Then we find that $\gamma_b^\star \approx 2 \cdot 10^{-3}$ and $\gamma_M^\star \approx 0.5 \cdot 10^{-3}$. Then, to confirm our results, we change the number of iterations to $K = 10^6$ and monitor the bias with the sequences of step sizes $\gamma_k = \gamma_b^\star k^{-\alpha}$ and the MSE with $\gamma_k = \gamma_M^\star k^{-\alpha}$ for several values of $\alpha$ in Figure S1. It can be observed that the optimal convergence rate is obtained for $\alpha = 1/3$ for the bias and $\alpha = 0.2$ for the MSE, which confirms the results of Theorem 2.

---

**Algorithm 3:** Stochastic Gradient Hamiltonian Monte Carlo with the Euler integrator.

---

**Input** : Initial state $\theta_0$, Step size $\gamma$, Parameter $\omega$, a probability distribution $\mathcal{L}$ of an unbiased estimators $\nabla\tilde{U}$ for $\nabla U$

**Output** : Samples $(\theta_k)_{k \geq 0}$

Initialize $r_0 \sim \mathcal{N}(0, \mathrm{Id})$,

**for** $k = 1, \cdots$ **do**

    Draw $Z_{k+1} \sim \mathcal{N}(0, \mathrm{Id})$

    Draw $\nabla\tilde{U}_k$ from $\mathcal{L}$

    $r_k = (1 - \omega\gamma)r_{k-1} - \gamma\nabla\tilde{U}_k(\theta_{k-1}) + \sqrt{2\omega\gamma}Z_k$

    $\theta_k = \theta_{k-1} + \gamma r_k$

---

**Algorithm 4:** Stochastic Gradient Richardson-Romberg Hamiltonian Monte Carlo with the Euler integrator.

---

**Input** : Random number seed $\mathcal{S}$, Step size $\gamma$, Parameter $\omega$, Initial state $\theta_0^{(\gamma/2)} = \theta_0^{(\gamma)} = \theta_0$, Test function $f(\cdot)$, Number of iterations $K$, a probability distribution $\mathcal{L}$ of an unbiased estimators $\nabla\tilde{U}$ for $\nabla U$

**Output** : $\hat{\pi}_K^{\mathrm{R}}(f) \approx \int f(\theta)\pi(d\theta)$

Initialize $r_0 \sim \mathcal{N}(0, \mathrm{Id})$, and set $r_0^{(\gamma)} = r_0^{(\gamma/2)} = r_0$

`// Run two chains in parallel with consistent Brownian increments`

**Chain 1:**

**for** $k = 1, \cdots, K$ **do**

   Set random number generator seed to $\mathcal{S}$

   Draw $Z_{2k-1}^{(\gamma/2)} \sim \mathcal{N}(0, \mathrm{Id})$, $Z_{2k}^{(\gamma/2)} \sim \mathcal{N}(0, \mathrm{Id})$

   Set $Z_k^{(\gamma)} = (Z_{2k-1}^{(\gamma/2)} + Z_{2k}^{(\gamma/2)})/\sqrt{2}$

   Draw $\nabla\tilde{U}_k^{(\gamma)}$ from $\mathcal{L}$

   $r_k^{(\gamma)} = (1 - \omega\gamma)r_{k-1}^{(\gamma)} - \gamma\nabla\tilde{U}_k^{(\gamma)}(\theta_{k-1}^{(\gamma)}) + \sqrt{2\omega\gamma}Z_k^{(\gamma)}$

   $\theta_k^{(\gamma)} = \theta_{k-1}^{(\gamma)} + \gamma r_k^{(\gamma)}$

Compute $\hat{\pi}_K^{(\gamma)}(f) = \frac{1}{K}\sum_{k=1}^{K} f(\theta_k^{(\gamma)})$

**Chain 2:**

**for** $k = 1, \cdots, 2K$ **do**

   Set random number generator seed to $\mathcal{S}$

   Draw $Z_k^{(\gamma/2)} \sim \mathcal{N}(0, \mathrm{Id})$

   Draw $\nabla\tilde{U}_k^{(\gamma/2)}$ from $\mathcal{L}$

   $r_k^{(\gamma/2)} = (1 - \frac{\omega\gamma}{2})r_{k-1}^{(\gamma/2)} - \frac{\gamma}{2}\nabla\tilde{U}_k^{(\gamma/2)}(\theta_{k-1}^{(\gamma/2)}) + \sqrt{\omega\gamma}Z_k^{(\gamma/2)}$

   $\theta_k^{(\gamma/2)} = \theta_{k-1}^{(\gamma/2)} + (\gamma/2)r_k^{(\gamma/2)}$

Compute $\hat{\pi}_K^{(\gamma/2)}(f) = \frac{1}{2K}\sum_{k=1}^{2K} f(\theta_k^{(\gamma/2)})$

`// RR extrapolation.`

$\hat{\pi}_K^{\mathrm{R}}(f) = 2\hat{\pi}_K^{(\gamma/2)}(f) - \hat{\pi}_K^{(\gamma)}(f)$

---

**Algorithm 5:** Stochastic Gradient Hamiltonian Monte Carlo with a symmetric splitting integrator.

---

**Input** : Initial state $\theta_0$, Step size $\gamma$, Parameter $\omega$, a probability distribution $\mathcal{L}$ of an unbiased estimators $\nabla\tilde{U}$ for $\nabla U$

**Output** : Samples $(\theta_k)_{k\geq 0}$

Initialize $r_0 \sim \mathcal{N}(0, \mathrm{Id})$,

**for** $k = 1, \cdots$ **do**

   $\theta_k^{(1)} = \theta_{k-1} + (\gamma/2)r_k$

   $r_k^{(1)} = \mathrm{e}^{-\omega\gamma/2}r_k$

   Draw $Z_{k+1} \sim \mathcal{N}(0, \mathrm{Id})$

   Draw $\nabla\tilde{U}_k$ from $\mathcal{L}$

   $r_k^{(2)} = r_k^{(1)} - \gamma\nabla\tilde{U}_k(\theta_k^{(1)}) + \sqrt{2\omega\gamma}Z_k$

   $r_k = \mathrm{e}^{-\omega\gamma/2}r_k^{(2)}$

   $\theta_k = \theta_k^{(1)} + (\gamma/2)r_k$

---

## Footnotes

[1]Note that they are not produced by a standard SGLD but the analysis is the same.