[Reviews · NeurIPS 2016]

Reviewer 1

Summary

This paper applies Richardson-Romberg extrapolation (RRE) to recent "big-data MCMC" methods, with a focus on Stochastic Gradient Langevin Dynamics (SGLD). RRE is a standard numerical integration tool that has recently seen some nice applications to numerical stochastic differential equations (SDEs); one of the nice features of this paper is that it provides a link between the MCMC literature and the SDE literature, which may be beneficial for both communities. The authors introduce a new SGLD-RRE method, develop some theoretical results on the convergence of the new algorithm (along with arguments for when SGLD-RRE improves on vanilla SGLD), and show some numerical results. They obtain impressive improvements on toy numerical examples, and modest but consistent improvements in large-scale examples. Overall it's a solid paper, clearly written.

Qualitative Assessment

As I said above, this is a solid paper, with interesting links between two somewhat disconnected fields; interesting theoretical results; and good empirical performance of the novel proposed algorithm. The paper is very clear and easy to read and seems technically sound. The method ends up being a fairly straightforward application of RRE to SGLD, which is computationally convenient. I believe practitioners will use these methods, though the methods will not revolutionize current practice. Minor comments: I found Fig 1a slightly misleading - this is not the empirical estimated posterior distribution here, but rather a gaussian with the empirical posterior mean and variance plugged in. This should be clarified / emphasized a bit more for the unwary reader. Fig 2a – it would be helpful to add a bit of discussion about why the bias is not necessarily decreasing as the stepsize decreases; this is a bit counter-intuitive.

Confidence in this Review

2-Confident (read it all; understood it all reasonably well)


Reviewer 2

Summary

This paper applies the Richardson-Romberg (RR) extrapolation to stochastic gradient Langevin dynamics in order to reduce the bias introduced from the gradient estimation noise. The resulting SGRRLD has an asymptotically lower bias and MSE than SGLD. The experiment with synthetic Linear Gaussian Model shows the advantage of the new algorithm over SGLD. Experiments on the MovieLens data show a slight improvement on the test RMSE but the results are arguable.

Qualitative Assessment

Richardson-Romberg (RR) extrapolation was shown to improve the convergence rate of Monte Carlo estimates on SDEs in the literature. This paper applies this technique to SG-MCMC algorithms and proves that under certain conditions the convergence rate of SGRRLD is faster than SGLD with the same step size at the price of running two parallel Markov chains. The algorithm description is clear and it is quite easy to implement the proposed algorithm. The synthetic experiments show the clear advantage of SGRRLD in reducing the bias in that particular problem. It would be very nice to see more real data experiments to show that a simple modification to SGLD will reduce the bias significantly. I have not checked the proof details of Theorem 1&2, but I’m curious that what condition is required for U to satisfy both theorems? Can U be multi-modal? Intuitively when the gradient noises are not correlated, the two chains will become uncorrelated quickly even if the Gaussian noises are correlated. How would the combination of the two chains reduces the order of the error. When run in real data experiments, do the authors observe the correlation between the two chains? My concern about the experiment setting and the real data experiment is that the authors argue that the computation time is about the same as SGLD with the parallelisation. However, we can also run SGLD with multiple parallel chains within the same time. If we compare the estimation from 2 chains of SLGD with the SGRRLD, can we still observe the same improvement in RMSE? Another problem is that in the real data experiment, we compare SGRRLD with a step size gamma and gamma/2 against SGLD with a step size gamma. I think we should also compare with SGLD with the step size gamma/2 that has a lower bias (and run 2 chains in parallel). Also, more real data experiments should be carried out.

Confidence in this Review

2-Confident (read it all; understood it all reasonably well)


Reviewer 3

Summary

Stochastic Gradient MCMC (SG-MCMC) methods provide an easy way to perform Bayesian posterior inference in large datasets, and have received a lot of interest from the research community in the last few years. One drawback of these methods is that they are asymptotically biased, meaning they do not converge exactly to the posterior distribution. This paper shows that the bias can be reduced by a technique called Richardson-Romberg extrapolation. The basic idea is very simple: Two SG-MCMC chains are run in parallel, with Chain 2 having half the step size of Chain 1. Also chain 2 is run for twice the number of steps as Chain 1. The injected Gaussian noise in the chains are correlated, with alpha_k = (beta_{2*k-1) + beta_{2*k})/sqrt(2) where alpha_k and beta_k are the injected Gaussian noise at time step k in Chains 1 and 2 respectively. Then, to compute the expectation of a function f w.r.t. the target distribution of the chains, one simply subtracts the estimate computed using samples from Chain 1 from the estimate computed using Chain 2. The resulting estimate is shown to have lower bias O(gamma^2) compared to O(gamma) obtained from Chain 1 or 2 alone (where gamma is the step size). They also prove a similar reduction in MSE of the estimates.

Qualitative Assessment

I like the fact that the method is very simple to understand and implement (see my summary), and does not require any major changes to the base SG-MCMC algorithm. Also, this seems very general and applies to a large class of SG-MCMC algorithms, and is therefore potentially very impactful to the Stochastic Gradient MCMC community. Novelty: Although Richardson-Romberg extrapolation is well known in numerical analysis, it is not widely known in the machine learning / stochastic gradient MCMC community. Clarity: The paper is well written and the presentation is clear. Comments/questions: - Can this technique be directly applied to all SG-MCMC methods? If not, are there specific conditions other than that the SG MCMC algorithm satisfies the bound in section 2.2: pi_gamma(f) = pi(f) + C * gamma + O(gamma^2)? - Can this method be applied to other non-stochastic gradient, but approximate MCMC schemes, e.g. sub-sampling based Metropolis-Hastings tests? What about Distributed SGLD? - Since one chain has to run for twice as long as the other, there is a lot of wasted computation where the chain with the bigger step size stays idle. Instead, is it possible to use 3 chains, one chain with a step size of gamma, and two chains with step sizes gamma/2 each, and run all of them for the same number of steps? Does this increase the variance of the estimate? - Is there an extension of this method to more than 2 step sizes? - The injected noise in the chains have to be correlated, but I didnt quite understand the effect of correlation between mini-batches . Is the variance lowest when using the same batch for the k^th iteration of chain 1, and the 2k-1^th and the 2k^th iteration of chain 2? Or is the effect of this correlation negligible? - I think the matrix factorization experiment (figure 5) should be run until convergence, and you should also compare to SGD. The current plot shows that your method works better than SGLD, but it would be nice to show that your method works better than SGLD in a setting where being Bayesian is advantageous (i.e. better than SGD). Bayesian matrix factorization is not widely used in practice, so showing some numbers will convince more people to adopt your method. Also having an SGD plot as control will put the final errors of SGRRLD and SGLD in better perspective. - Although the method is easy and clearly presented, an algorithm box could make the paper more accessible to a wider audience. I believe many practitioners will not read through the more theoretical parts, so pseudo code presented early on in the paper will enable readers who are not very familiar with this area to try out your method without understanding the more challenging parts.

Confidence in this Review

2-Confident (read it all; understood it all reasonably well)


Reviewer 4

Summary

This paper improves existing SG-MCMC by proposing a Richardson-Romberg scheme, such that it improves the rates of convergence in terms of bias and MSE. The scheme is easy to implement, which requires minimal modifications to traditional SG-MCMC. The paper is also supported by solid theoretical results.

Qualitative Assessment

The paper proposes a Romberg-Richardson scheme that can improve the convergence rates of standard SG-MCMC algorithms, the method looks nice and theory looks solid. The most concerned part is about how the specific convergence rates are obtained from the theorem (as explained in the "explain fatal flaws" section. Except for this, I also have a practical concern about the matrix factorization experiment result. To be fair, if we run the proposed method on a single core computer, it would take 3 time longer than standard SG-MCMC because it has two chains and one of them uses two times more steps. If we take this into account, the plots in Figure 5 would not be fair, the time for SGLD should be multiplied by a factor of 2/3, which would shrink the difference between SGLD and SGRRLD. In Figure 3, the bias plot, it seems the best convergence is achieved by the red cure, which corresponds to \alpha=0.2, not 1/3 stated in the text, anything wrong? Also, the optimal curve looks fluctuating a lot, thus the rate would not match \alpha=0.2, any reasons why this happens? Also, it is stated in line 256 when stepsize is too small, the term 1/K\gamma would dominate. I wonder why this does not happen for SGLD? Some minor comments: Eq.1, why do you use a different symbol than \theta? line 94, RR not defined. line 131, \Gamma_K not defined. line 160, smooth function f, what kinds of smoothness? line 249: Figure 1(b) plots Bias vs. time, not dimension, should it be Figure 1(c)? Also, please use bold letters to represent *vectors*.

Confidence in this Review

3-Expert (read the paper in detail, know the area, quite certain of my opinion)


Reviewer 5

Summary

To reduce the bias of SG-MCMC, the authors propose a novel method called stochastic gradient Richardson-Romberg MCMC. This method employs Richardson-Romberg extrapolation, an off-the-shelf tool for SDE. In particular, two chains are run embarrassingly parallel in SG-RR-MCMC without communication. The theoretical analysis is provided, showing that SG-RR-MCMC is asymptotically consistent. Empirical results are also satisfactory.

Qualitative Assessment

The idea is interesting and proved to be effective according to empirical results. It can be seen as a simple combination of Richardson-Romberg extrapolation and SGLD. The efficiency can be validated theoretically via providing a better convergence rate, though this can be reached using high-order integrator (unfortunately, the authors didn't provide a comparison with the integrator in experiment). However, there exists inconsistent statement about the step size as mentioned above. Furthermore, Algorithm 1 (in supplement) is not consistent with Equation (5).

Confidence in this Review

2-Confident (read it all; understood it all reasonably well)


Reviewer 6

Summary

This paper proposes to use a numerical sequence acceleration method (higher-order numerical integrator), the Richardson-Romberg extrapolation, to improve the convergence of SG-MCMC algorithm. It requires running two SG-MCMC chains in parallel, with different step sizes and correlated injecting Gaussian noise. The convergence of both asymptotic and non-asymptotic properties are provided. The experiments on synthetic Gaussian model verify the bias and MSE bounds of theorems, and the experiments of large-scale matrix factorization on real data demonstrate of the practical use of proposed algorithms over standard SGLD.

Qualitative Assessment

This is a nice paper overall in terms of proposed technique and presentation. I feel that this paper proposes a valid contribution to the area of SGMCMC methods, and does a good job putting this method in context with similar previous methods. I have two main concerns: (1) The novelty of the paper may be incremental, given that the effect of numerical integrator on SGMCMC has been analyzed in [10], and proposed RR extrapolation has been used for approximation of invariant distributions in [23]. (2) The pracitcal use of algorithm can be limited, and there may be a lack of fair comparison in experiments. This is parallel implementation of SG-MCMC algorithm, and the fact of using correlated Brownian motion requires the two chains to communicate at every step. However, it seems the authors did not consider the communication cost, assume there is no cost. It is better to mention this. It is not clear how the computing resource is assigned for the SGRRLD and SGLD in experiment. For example, is the single chain of SGLD implemented using the resouce that the two chains of SGRRLD use? If not, this may be not a fair comparison for pratical consideration. If yes, then question is how to split the resource for the two chains of SGRRLD in the algorithm procedure. The computation cost of SGMCMC mostly comes from computing stochastic gradient, I would wonder how could partial resource used in SGRRLD for computing gradient at each chain lead to less wall-clock time, than full resource used in SGLD? Please add more details on the setup of implementation. Also, the alternative is to run two chains of SGLD embarrassingly (without communication), is it possible to compare with this? Minor comments: (1) "Unadjusted Langevin Algorithm (ULA)" appears twice, line 25 and 70.

Confidence in this Review

2-Confident (read it all; understood it all reasonably well)